# RagD auto-activating mutations impair MiT/TFE activity in kidney tubulopathy and cardiomyopathy syndrome

Irene Sambri [1,2,14], Marco Ferniani [1,2,14], Giulia Campostrini [3], Marialuisa Testa[1], Viviana Meraviglia [3], Mariana E. G. de Araujo [4], Ladislav Dokládal[5], Claudia Vilardo [1], Jlenia Monfregola[1], Nicolina Zampelli[1], Francesca Del Vecchio Blanco [6], Annalaura Torella[1,6], Carolina Ruosi[7], Simona Fecarotta[2], Giancarlo Parenti[1,2], Leopoldo Staiano [1,8], Milena Bellin [3,9,10], Lukas A. Huber [4], Claudio De Virgilio [5], Francesco Trepiccione[7,11], Vincenzo Nigro[1,6] & Andrea Ballabio [1,2,12,13] ✉

Heterozygous mutations in the gene encoding RagD GTPase were shown to cause a novel autosomal dominant condition characterized by kidney tubulopathy and cardiomyopathy. We previously demonstrated that RagD, and its paralogue RagC, mediate a non-canonical mTORC1 signaling pathway that inhibits the activity of TFEB and TFE3, transcription factors of the MiT/TFE family and master regulators of lysosomal biogenesis and autophagy. Here we show that RagD mutations causing kidney tubulopathy and cardiomyopathy are "auto-activating", even in the absence of Folliculin, the GAP responsible for RagC/D activation, and cause constitutive phosphorylation of TFEB and TFE3 by mTORC1, without affecting the phosphorylation of "canonical" mTORC1 substrates, such as S6K. By using HeLa and HK-2 cell lines, human induced pluripotent stem cell-derived cardiomyocytes and patient-derived primary fibroblasts, we show that *RRAGD* auto-activating mutations lead to inhibition of TFEB and TFE3 nuclear translocation and transcriptional activity, which impairs the response to lysosomal and mitochondrial injury. These data suggest that inhibition of MiT/TFE factors plays a key role in kidney tubulopathy and cardiomyopathy syndrome.

Inherited kidney tubulopathies are kidney homeostasis disorders caused by dysfunctional proteins involved, either directly or indirectly, in the tubular transport of water and solutes. This group of diseases is highly heterogeneous, both genetically and clinically[1,2]. A recent study described nine families with an autosomal dominant disease entity characterized by the association of kidney tubulopathy and hypomagnesemia with severe dilated cardiomyopathy[3]. All affected individuals in these families carried mutations in the *RRAGD* gene (also named *RAGD*), encoding the GTPase RagD[3]. Rag GTPases, heterodimeric complexes formed by RagA or B bound to RagC or D, are involved in the activation of the mechanistic Target Of Rapamycin Complex 1 (mTORC1) by mediating its recruitment to the lysosomal surface[4–8]. To activate mTORC1, RagA/B must be in the GTP-bound state, whereas the nucleotide-binding state of RagC/D does not play a major role in mTORC1-mediated phosphorylation of "canonical" substrates, such as S6K and 4E-BP1[4–9]. By contrast, we showed recently that GDP-binding of RagC/D, which is driven by the GTPase-activating protein (GAP) folliculin (FLCN), plays a crucial

---

role in mTORC1-mediated selective phosphorylation and cytoplasmic retention of the Transcription Factors EB and E3 (TFEB and TFE3)[4], master controllers of lysosomal biogenesis and autophagy[10,11]. Consistent with this, loss of function of folliculin (FLCN) leads to constitutive nuclear localization and activation of TFEB and TFE3, without affecting the activity of mTORC1 on other substrates such as S6K[4,12,13]. This mTORC1 substrate-specific pathway was named "non-canonical mTORC1 signaling"[14]. A recent study determined the Cryo-EM structure of the mTORC1-TFEB-Rag-Ragulator complex supporting the presence of both "canonical" and "non-canonical" branches of the pathway[15]. Here we describe a new family with kidney tubulopathy and cardiomyopathy syndrome carrying a novel RRAGD mutation (P88L) that, similarly to previously described mutations[3], impairs RagD ability to bind GTP, leading to a constitutive active protein. This leads to constitutive phosphorylation of TFEB and TFE3, thus inhibiting their nuclear translocation both in FLCN KO and in wild-type cells subjected to lysosomal or mitochondrial stress conditions. These observations suggest that TFEB and TFE3 inhibition drives kidney tubulopathy

and dilated cardiomyopathy in patients with RagD auto-activating mutations.

## Results

### Identification of a novel *RRAGD* mutation

We identified a large family affected by hypomagnesemia, mild hypokalemia and severe medullary nephrocalcinosis associated with heart disease including arrhythmias, valvulopathies, myocardial infarction and dilated cardiomyopathy. Clinical and laboratory data of this family are reported in Fig. 1a, b, Supplementary Table 1 and in the Supplementary data section. We collected data from a wide range of ages (from 5 to 62 y/o) covering three generations (Fig. 1c) and performed whole-exome sequencing analysis which revealed that affected members carried a novel c.263 C > T (p.P88L) mutation in the *RRAGD* gene. Similarly to previously described families with kidney tubulopathy and cardiomyopathy[3], the mutated amino acid, proline 88, is highly conserved and located within the RagD GTP-binding motif at the N-terminus of the protein[16] (Fig. 1d).

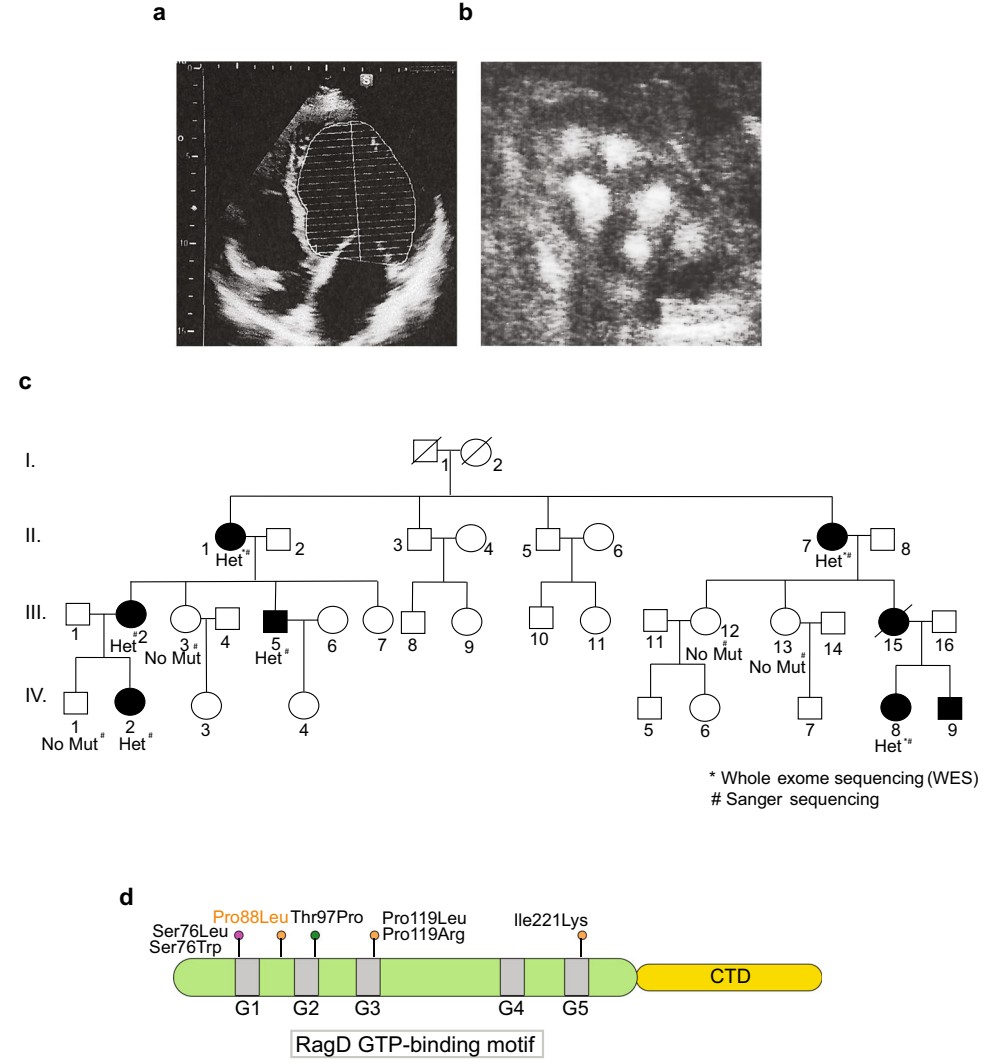

**Fig. 1 | A new *RRAGD* mutation associated with tubulopathy and cardiomyopathy. a** Representative image of left ventricle ultrasound from patient II.1, showing ventricle dilatation (Left ventricular end-diastolic diameter 62 mm and area 42,02 cm²). **b** Representative image of kidney ultrasound of patient III.15; showing severe medullary nephrocalcinosis. **c** Pedigree of the family and indication of the individuals carrying the p.P88L *RRAGD* mutation. Filled symbols represent affected individuals carrying the mutation and symptomatic for kidney and/or heart abnormalities. **d** Domain organization of RagD protein with GTP-binding motifs reported in light gray (G1–G5). Pathogenic mutations affecting RagD protein stability, binding to GTP, and Mg²⁺ coordination are indicated in orange, green, and magenta, respectively.

## In silico modeling and in vitro assays reveal a gain of function of RagD mutations

To evaluate the molecular consequences of the P88L mutation, as well as of the previously described *RRAGD* mutations[3], we predicted computationally if these would alter protein stability and/or binding to either magnesium or the phosphate (Supplementary Tables 2 and 3). In addition, we generated in silico models of the mutations (Supplementary Fig. 1a–d). These analyses indicated that S76L and S76W mutations are likely to affect magnesium coordination (Supplementary Fig. 1b), which is required for GTP binding[17], T97P may directly affect GTP binding (Supplementary Fig. 1c), whereas the remaining mutations (P88L, P119L, P119R, and I221K) were predicted to decrease protein stability (Supplementary Fig. 1d). Switch I and Switch II of the Rag GTPases are likely to undergo major conformation changes during the activation cycle. Because of the lack of information on the RagD-GDP structure, we analyzed the corresponding residues in RagC to determine whether the effects of mutations on the stability of RagD were associated with the GTP-bound state (Supplementary Table 2). Interestingly, the disease-causing mutations did not seem to alter RagC/D stability in the nucleotide-free or GDP-bound states. Thus, all these mutations are predicted to affect, either directly or indirectly, GTP loading. To test this prediction, we generated recombinant proteins for the WT and mutated RagD versions and tested them for GTP-binding and GTPase activity assays (see "Methods"). WT RagD bound GTP in a concentration-dependent manner, whereas all RagD mutants were impaired in GTP binding, irrespective of the protein concentration (Supplementary Fig. 1e, f). Moreover, all the disease-associated mutations impaired the GTP hydrolysis activity of RagD (Supplementary Fig. 1g, h). These data indicate that RagD mutants are unable to bind GTP and are thus in a nucleotide-free state or a GDP-loaded state, which are both active conformations, indicating that these are auto-activating mutations.

In a previous study, we showed that only active (i.e., GDP-loaded) RagC/D physically interact with TFEB, thus mediating a novel substrate recruitment mechanism that enables mTORC1 to phosphorylate TFEB[4,14]. Indeed, RagC/D interaction with TFEB is lost in *FLCN* KO cells due to constitutive inactivation of RagC/D[4,14]. To test if the RagD auto-activating mutations identified in Schlingmann et al.[3] and in the present study were able to induce the interaction with TFEB in a FLCN independent-manner, we performed co-immunoprecipitation (Co-IP) analysis in HeLa *FLCN* KO cells transiently expressing TFEB and Rag GTPases (RagD WT or RagD mutants in combination with RagA WT or RagB WT). The results showed that RagD disease-associated mutations, unlike RagD wild type, rescued the interaction between RagD and exogenous TFEB in *FLCN* KO cells (Fig. 2a and Supplementary Fig. 1i, j). A similar effect was observed when probing the interaction between endogenous TFEB and transiently expressed RagD mutants (Fig. 2b). Interestingly, this interaction occurred in an mTORC1-independent manner (also in the presence of Torin) (Fig. 2b). Overall, these mutations cause RagD auto-activation with no requirement of FLCN GAP activity.

## RagD auto-activating mutants inhibit TFEB/3 activity

Consistent with their ability to induce RagD-TFEB interaction in *FLCN KO* cells, we found that mutant RagD cDNAs rescued mTORC1-mediated TFEB phosphorylation in HeLa *FLCN* KO cells, as detected both by the analysis of a molecular weight shift of TFEB (Fig. 2c and Supplementary Fig. 2a) and using a phospho-antibody against TFEB (Serine 211)[18] (Supplementary Fig. 2c, d). As expected, Torin treatment prevented this effect (Fig. 2c), indicating that RagD mutations induce TFEB phosphorylation by mTORC1. Similar results were obtained using dermal fibroblasts generated from a patient carrying the *RRAGD* P88L mutation (Fig. 2d). Importantly, the expression of RagD mutants had no effect on the phosphorylation of the canonical mTORC1 substrate S6K (Fig. 2c, d and Supplementary Fig. 2a, b). We then evaluated the

lysosomal recruitment-detachment of mTORC1 in the presence of RagD mutants either under basal or amino acid starvation conditions. HeLa cells expressing WT or mutant RagD showed the expected detachment of mTORC1 from lysosomes in amino acid starved cells (Supplementary Fig. 2e, f). Similar results were obtained in patient-derived fibroblasts carrying the *RRAGD* P88L mutation (Supplementary Fig. 2g). In contrast, HeLa cells expressing RagA Q66L, a mutated version of RagA, that is known to promote mTORC1 lysosomal recruitment and activity towards S6K[19] during amino acid starvation, showed constitutive mTORC1 lysosomal localization (Supplementary Fig. 2e, f). Taken together, these results demonstrate that RagD mutations promote non-canonical mTORC1 signaling[14], leading to TFEB phosphorylation without affecting mTORC1 lysosomal localization and its activity towards canonical mTORC1 substrates. Interestingly, the expression of RagD mutants leads to an increase of both exogenous and endogenous TFEB protein levels in HeLa *FLCN* KO cells (Fig. 2a, b), possibly by enhancing TFEB stability, through an increased interaction with 14-3-3, only in the presence of RagD mutants (Supplementary Fig. 2h). Otherwise, we do not appreciate the same increase of TFEB protein levels in fibroblast derived-patient carrying the P88L heterozygous mutation, probably due to the different conditions between the two cell lines.

Based on these observations, we tested whether the effect of RagD mutants on TFEB phosphorylation would also affect the subcellular localization of TFEB. Expression of RagD cDNAs carrying disease-causing mutations P88L, S76L, T97P, P119L, and I221K in HeLa *FLCN* KO cells, promoted TFEB cytoplasmic re-localization (Fig. 2e, f and Supplementary Fig. 3a). Similar results were obtained in HK-2 *FLCN* KO cells carrying an inducible TFEB-GFP system (Fig. 2g, h and Supplementary Fig. 3b), as well as in patient-derived fibroblasts carrying the *RRAGD* P88L mutation (Fig. 2i, j). As expected, TFE3 behaved like TFEB, consistent with the evidence that the nucleo-cytoplasmic shuttling of these transcription factors is regulated by the same mechanisms[20–23] (Supplementary Fig. 3c). Finally, to monitor whether RagD mutant-induced TFEB cytoplasmic re-localization resulted in inactivation of its transcriptional activity, we generated a new MiT-TFE transcriptional reporter by placing the NUC-mCherry fluorophore downstream of the promoter of *GPNMB* (GPNMBprom-NUC-mCherry), a highly sensitive TFEB transcriptional target[24,25]. A HeLa *FLCN* KO cell line stably expressing the GPNMBprom-NUC-mCherry reporter was transiently transfected with HA-tagged RagD mutants. A significant decrease of reporter activity was observed in the presence of RagD mutants indicating that these mutations inhibit TFEB transcriptional activity by inducing its cytoplasmic localization (Supplementary Fig. 3d, e). Consistent with these data, Real Time-PCR analysis showed marked reduction of the expression of TFEB target genes in HeLa *FLCN* KO cells transfected with RagD mutants compared to cells transfected with WT RagD (Supplementary Fig. 3f). Moreover, performing the same experiment in TFEB/TFE3 knockdown cells, we confirmed that the inhibition of the expression of target genes is correlated with the expression of TFEB and TFE3 (Supplementary Fig. 3g). Together, these data strongly suggest that RagD auto-activating mutations have a strong impact on kidney tubulopathy and cardiomyopathy by inhibiting MiT-TFE nuclear translocation and activity.

## RagD mutations affect TFEB localization and activity during lysosomal and mitochondrial injury

Both TFEB and TFE3 are key regulators of the lysosomal-autophagic pathway[10,11,20,22,26], a crucial process that controls cellular homeostasis and survival[27], which is defective in several diseases[28]. TFEB is known to promote intracellular clearance through its transcriptional control of the lysosomal-autophagic pathway[29]. Indeed, TFEB overexpression improves the phenotypes associated with various diseases characterized by autophagy defects, including kidney diseases and cardiac hypertrophy[29–31]. Recently, it has been shown that treatments

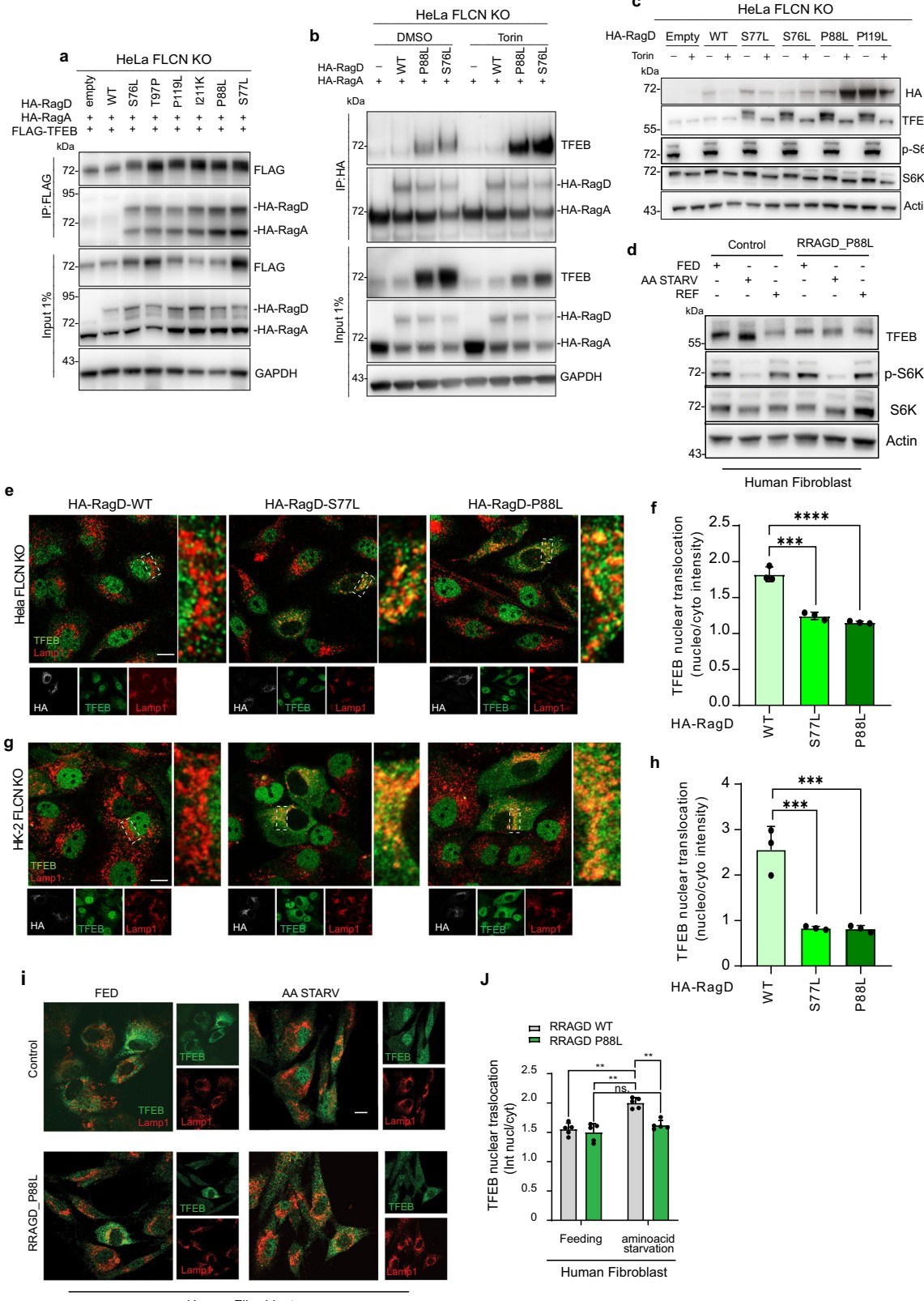

with the lysosomotropic compound l-leucyl-l-leucine methyl ester (LLOMe) or with the TRPML1 agonist MK6-83 selectively inhibit the phosphorylation of TFEB by mTORC1, thus promoting its nuclear translocation, without affecting the phosphorylation of the canonical mTORC1 substrates[32,33]. TFEB activation during lysosomal damage is essential for the maintenance of lysosome homeostasis[33]. Remarkably,

in cells transfected with auto-activating RagD mutants, treatment with LLOMe (Fig. 3a, b and Supplementary Fig. 4a, b) or with MK6-83 (Fig. 3d, e and Supplementary Fig. 4c, d) failed to induce the nuclear translocation of TFEB, which was retained in the cytosol and phosphorylated by mTORC1 on S211 (Supplementary Fig. 4e). Furthermore, neither of the treatments altered the phosphorylation of S6K, a

**Fig. 2 | RagD mutations impair TFEB nuclear translocation.**
**a** Immunoprecipitation performed in HeLa *FLCN* KO transfected with FLAG-TFEB (bait) and HA-RagD WT or HA-RagD mutants (S76L, T97P, P119L, I221K, P88L, S77L) and with equimolar amount of HA-RagA WT (*n* = 2 independent experiments).
**b** Immunoprecipitation performed in HeLa *FLCN* KO transfected with HA-RagD WT or HA-RagD mutants (P88L and S76L) and with an equimolar amount of HA-RagA WT to analyze the interaction with the endogenous TFEB. Cells were also treated with Torin (*n* = 2 independent experiments). **c** Western blot of cell lysates of HeLa *FLCN* KO cells transiently transfected with HA-RagD WT or HA-RagD mutants (S77L, S76L, P88L, P119L) and treated with Torin. anti-TFEB, anti-phopho-S6K, anti-S6K, and anti-actin antibodies were used (*n* = 3 independent experiments). **d** Western blot representing TFEB molecular weight shift, phospho-S6K and total S6K in human fibroblasts carrying the *RRAGD* P88L mutation and relative control fibroblasts. Cells were analyzed in normal feeding (FED), amino acid starved (AA STARV) and amino acid replenished conditions (REF) (*n* = 2 independent experiments).
**e** Representative images of HeLa *FLCN* KO cells transiently transfected with HA-RagD-WT or HA-RagD mutants (S77L and P88L) and immunostained with anti-TFEB, anti-LAMP1 and anti-HA antibodies. Scale bar, 10 μm. **f** Graph shows TFEB nuclear-

cytosolic ratio intensity in HeLa *FLCN* KO cells transiently transfected with HA-RagD-WT or HA-RagD mutants (S77L, P88L) (mean ± s.d. of *n* > 1000 cells from *n* = 3 independent experiments). Quantification performed using the Perkin-Elmer Opera system (see "Methods"). Ordinary one-way ANOVA Tukey multiple comparison test (***$p < 0.001$, ****$p < 0.0001$). **g** HK-2 *FLCN* KO cells expressing TFEB-GFP were transiently transfected with HA-RagD-WT or HA-RagD mutants (S77L and P88L) and immunostained with anti-HA and anti-LAMP1 antibodies. Scale bar, 10 μm. **h** Graph shows TFEB nuclear-cytosolic ratio intensity in HK-2 *FLCN* KO TFEB-GFP transiently transfected with HA-RagD-WT or HA-RagD mutants (S77L, P88L) (mean ± s.d. of *n* > 1000 cells from n = 3 independent experiments). Quantification performed as in (**f**). Ordinary one-way ANOVA Tukey multiple comparison test (***$p < 0.001$). **i** Representative immunofluorescence images of human fibroblast from a patient with *RRAGD* P88L and from a control patient in normal feeding (FED) and upon amino acid starvation (AA STARV). Cells were stained with anti-TFEB and anti-LAMP1. Scale bar, 10 μm. **j** Graph shows TFEB nuclear-cytosolic ratio intensity during normal feeding and amino acids starved conditions (mean ± s.d. of *n* > 1000 cells from *n* = 3 independent experiments). Quantification performed as in (**f**). Ordinary two-way ANOVA Sidak's multiple comparison test (ns > 0.9, **$p < 0.01$).

canonical mTORC1 substrate, in cells overexpressing RagD mutants (Fig. 3c, f). Finally, upon LLOMe or MK6-83 treatment, the expression of TFEB target genes was blunted in RagD mutant-expressing cells (Supplementary Fig. 4f, g), indicating a specific impairment of the TFEB-mediated response to lysosomal damage.

To confirm these data in a more physiological and relevant cellular model, we generated human iPSC-derived cardiomyocytes (hiPSC-CMs). Similarly to patient-derived fibroblasts, hiPSC-CMs expressing RagD mutants showed impaired TFEB nuclear translocation both during amino acid starvation (Supplementary Fig. 5a) and upon treatment with LLOMe (Fig. 3g, h and Supplementary Fig. 5b), thus confirming the results obtained in HeLa and HK-2 cells.

Activation of TFEB during mitophagy through a PINK1- and Parkin-dependent mechanism[34] is pivotal for kidney resistance to stress stimuli[35], as well as for mitochondrial quality control in cardiomyocytes under stress[36]. The expression of RagD mutants in HK-2 renal cells treated with mitophagy-inducing agents, such as the mitochondrial electron transport chain inhibitors oligomycin and antimycin A (oligomycin/antimycin A [O/A]), which are known to trigger TFEB nuclear translocation[34], led to TFEB cytoplasmic retention (Fig. 4a, b and Supplementary Fig. 5c, d). In addition, we observed a reduction of mitophagy activation, as demonstrated by decreased Parkin mitochondrial translocation (Fig. 4c, d) and reduced PINK1 and Parkin protein levels (Fig. 4e–g). This suggests that RagD mutations identified in patients with kidney tubulopathy and cardiomyopathy impair TFEB-mediated mitophagy activation.

## Discussion

We identified a novel family with multiple individuals affected by kidney tubulopathy and cardiomyopathy syndrome and carry a c.263 C > T (P88L) mutation in the *RRAGD* gene. Similarly to previously described families with this disease[3], the disease-causing mutation is located in the nucleotide-binding region of RagD. In silico modeling of the RagD mutant identified in this family, as well as the mutations found in all previously described families, indicated that they suppress, either directly or indirectly, GTP binding to RagD as demonstrated by GTP binding and GTPase activity assays. Consequently, all RagD mutated proteins are preferentially in the nucleotide-free state or GDP conformation state, leading to constitutive activation of the protein.

In the original study, in which kidney tubulopathy and cardiomyopathy was first described by Schlingmann et al., the authors claimed that the disease was caused by mTORC1 hyperactivation induced by *RRAGD* mutations[3]. However, in that study the degree of mTORC1 hyperactivity induced by *RRAGD* mutations (as measured by S6K phosphorylation) was marginal and the effects were highly variable among the different mutations. Most importantly, the

phosphorylation of S6K in samples transfected with RagD mutants was still turned off by starvation, similarly to control sample, whereas in the sample transfected with active RagA mutant starvation had no effect (see Fig. 4B of ref. 3). These results suggest that RagD mutants do not cause constitutive mTORC1-mediated phosphorylation of S6K. Consistent with these findings, in the present study we show that RagD mutants induce constitutive phosphorylation of TFEB and other MiT-TFE factors, whereas they have no effect on mTORC1-mediated phosphorylation of S6K. The ability of RagD mutants to induce phosphorylation of TFEB without affecting the phosphorylation of S6K is in line with recent studies, including our own, that demonstrated the presence of an mTORC1 substrate-specific pathway, which we named "non-canonical mTORC1 signaling"[4,12–14,32,33]. Remarkably, in a recent study we provided structural evidence of mTORC1 substrate specificity[15].

Importantly, constitutive phosphorylation of MiT/TFE factors induced by *RRAGD* mutations leads to their cytoplasmic retention, strongly suggesting that MiT/TFE inhibition is the main mechanism underlying kidney tubulopathy and cardiomyopathy.

Moreover, these mutations also inhibited the lysosomal/autophagic response to lysosomal injury (LLOMe and MK6-83) and mitochondrial stress (O/A), underlining their biological relevance. A recent study showed that TFEB-mediated autophagic response plays a crucial role in CaOX-induced kidney damage[33]. Kidney stones, in particular calcium oxalate (CaOx), are associated with oxidative stress, inflammation, and tissue injury[37,38]. Mitochondria are critical for normal kidney function as they provide energy support to maintain ion homeostasis and eliminate waste metabolites and environmental toxicants, including drugs[39]. These data suggest that the kidney tubulopathy observed in patients carrying *RRAGD* mutations may be due to an impairment of the TFEB/TFE3-mediated autophagic response.

Moreover, the role of mitochondria is of particular importance in the heart as they provide ATP through oxidative phosphorylation to sustain contractile function. In pathological conditions, mitochondria also represent the primary source of reactive oxygen species that promote cardiomyocyte death and heart failure[40]. To protect against mitochondrial damage, cardiomyocytes use well-coordinated quality control mechanisms that maintain overall mitochondrial health through mitochondrial biogenesis, mitochondrial dynamics, and mitophagy[40]. Importantly, these quality control mechanisms are regulated at the transcriptional level by TFEB[41]. Recent studies showed that TFEB improves cardiomyocyte survival in several pathological conditions. Adeno Associated Viral vectors (AVV)-mediated TFEB overexpression was found to attenuate autophagic blockade, cardiomyocyte death, and heart failure in MAO-A transgenic mice[42]. Furthermore, it has been reported that loss of TFEB leads to cardiac

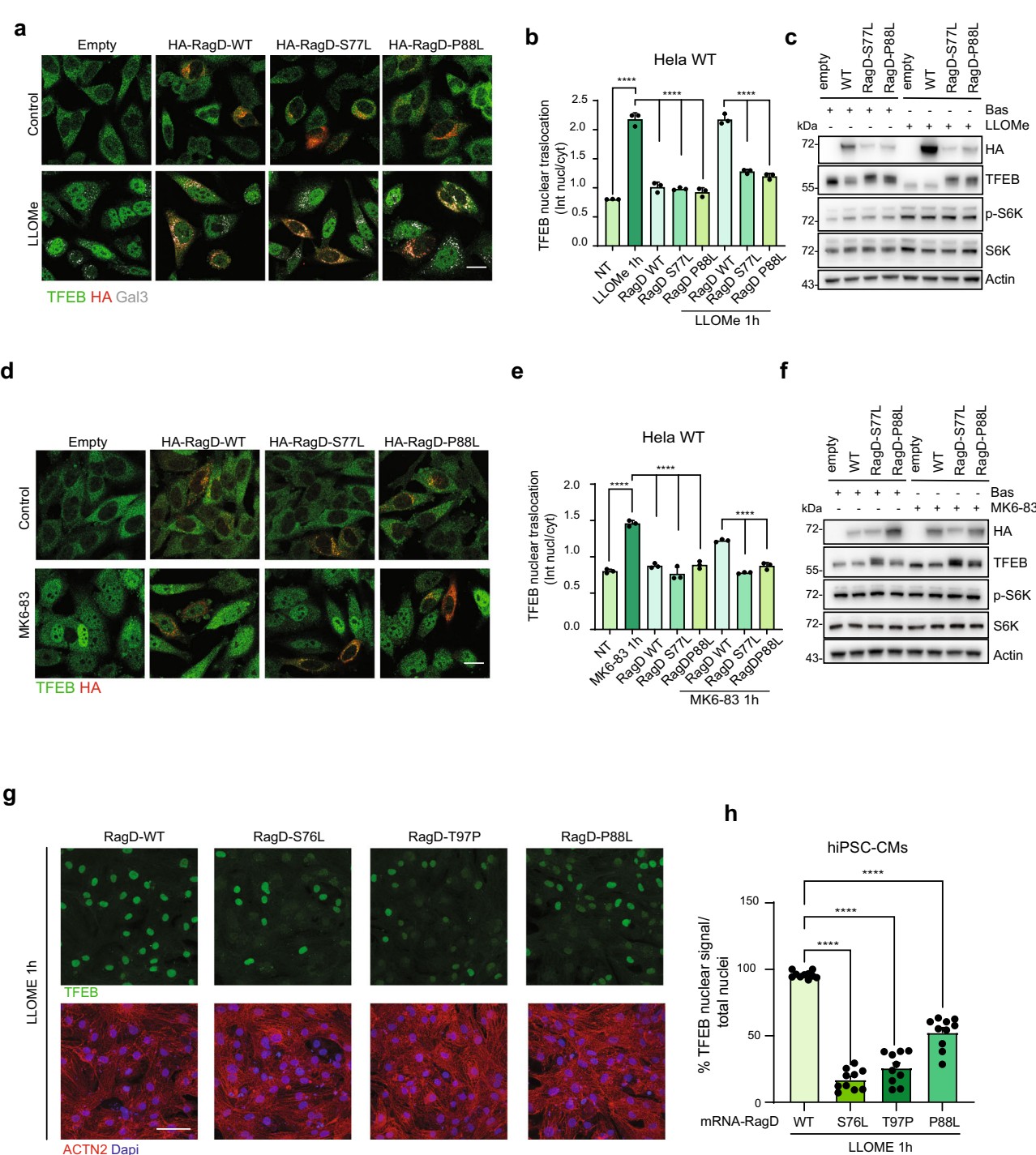

hypertrophy by blocking autophagic degradation of GATA4, a transcription factor responsible for the upregulation of several cardiac-specific fetal genes involved in cardiomyocyte growth[43]. Most importantly, a previous study described a de novo S75Y gain-of-function mutation in *RRAGC*, a *RRAGD* paralogue, in a fetus with syndromic fetal dilated cardiomyopathy (DCM)[44]. The same *RRAGC* mutation was introduced in a zebrafish model, which showed severe cardiomyopathy. Remarkably, the phenotype was rescued by TFEB overexpression[31].

Despite the beneficial effect of TFEB in the context of *RRAGC* auto-activating mutations[31], it has been shown that heart-specific TFEB overexpression induces cardiomyopathy in transgenic mice[45]. On the

other hand, AAV-mediated moderate TFEB overexpression seems to exert beneficial effects[46]. These studies suggest TFEB levels and sub-cellular localization may play a beneficial or detrimental role depending on the physio-pathological context.

In conclusion, our data suggest that constitutive activation of RagD causes inhibition of MiT/TFE activity, which causes defective cellular metabolic responses to different stimuli, such as lysosomal or mitochondrial damage. This can cause a cascade of pathological changes that ultimately contribute to the onset of kidney tubulopathy and cardiomyopathy and may be involved in additional disease enti-ties. Currently, many different types of renal tubulopathies and car-diomyopathies are considered idiopathic[1]. Our data suggest these

**Fig. 3 | RagD mutants impair TFEB response to lysosomal damaging agents.**
**a** Representative immunofluorescence images of HeLa cells transfected with HA-RagD WT or mutants (S77L and P88L) and treated with 500 μM LLOMe (for 1 h). Cells were stained with anti-TFEB and anti-HA. Staining for Galectin-3 (Gal3) was shown to visualize lysosomal membrane permeabilization. Scale bar, 10 μm. **b** Graph shows the TFEB nuclear-cytosolic ratio intensity of the HA positive (RagD WT or mutants S77L, P88L) cells treated with LLOMe (mean ± s.d. of $n > 1000$ cells from $n = 3$ independent experiments). Ordinary one-way ANOVA Tukey multiple comparison test (****$p < 0.0001$). Quantification performed as in Fig. 2f. **c** Western blot representing TFEB molecular weight shift, phospho-S6K and total S6K in HeLa cells transfected with HA-RagD WT or mutants (S77L and P88L) followed by LLOMe treatment ($n = 3$ independent experiments). **d** Representative immunofluorescence images of HeLa cells transfected with Ha-RagD WT or mutants (S77L and P88L) and treated with 30 μM MK6-83, a specific TRPML1 agonist, for 1 h. Cells were stained with anti-TFEB and anti-HA. Scale bar, 10 μm. **e** Graph shows the TFEB nuclear-cytosolic ratio intensity of the HA positive (RagD WT or mutants S77L, P88L) cells treated with MK6-83 (mean ± s.d. of $n > 1000$ cells from $n = 3$ independent experiments). Quantification performed as in Fig. 2f. Ordinary one-way ANOVA Tukey multiple comparison test (****$p < 0.0001$). **f** Western blot representing TFEB molecular weight shift, phospho-S6K and total S6K in HeLa cells transfected with HA-RagD WT or mutants (S77L and P88L) followed by MK6-83 treatment ($n = 3$ independent experiments). **g** Representative immunofluorescence staining for TFEB and sarcomeric marker ACTN2 in hiPSC-CMs transfected with mRNA-RagD-WT or -RagD mutants S76L, P88L, and T97P. Nuclei are stained with DAPI. Scale bar, 50 μm. **h** Graph showing the percentage of cells with TFEB nuclear signal/total nuclei in hiPSC-CMs transfected with mRNA-RagD-WT or -RagD mutants (S76L, P88L, T97P). $n = 10$ areas from 2 independent wells; one-way ANOVA followed by Tukey's multiple comparisons test. Data are shown as mean ± SEM (****$p < 0.0001$).

"idiopathic" forms may be caused by mutations in genes, such as *RRAGD*, involved in the regulation of the non-canonical mTORC1 pathway. Restoring MiT/TFE function, either by selectively inhibiting their phosphorylation or by promoting their nuclear translocation, may represent an effective therapeutic strategy for these conditions.

## Methods
The methods used in this research comply with all ethical regulations. In particular, for human samples, data collection and genetic analysis were performed after obtaining informed consent to participate and publish identifiable medical information from all the study participants. For children, written informed consent to participate and publish medical information was obtained from the legal guardian. For isolation of patient-derived fibroblasts each patient signed an informed consent before enrolling in this study. The study protocol covering all of the work involving humans in this manuscript has been approved by the Ethical Committee of the AUO policlinico Unicampania "Luigi Vanvitelli" (with the protocol number 7539).

### Materials
The reagents used in this study were obtained from the following sources: antibodies against phospho-p70 S6 kinase (Thr389) (1A5) (cat. # 9206 − 1:1000), p70 S6 kinase (cat. # 9202 − 1:1000), human TFEB (cat. # 4240 − 1:1000 WB/1:100 IF), TFEB-pS211 (E9S8N) (cat. # 37681 used at 1:1000 WB/1:100 IF), Parkin (cat # 2132 1:500 WB/1:200 IF); PINK1 (D8G3) (cat # 6946 1:1000 WB), mTOR (7C10) (cat # 2983 1:200 IF), GPNMB (E4D7P) XP (cat #38313 1:1000 WB) were from Cell Signaling Technology; antibodies against GAPDH (6C5) (cat. no. sc-32233-1:15000 WB), LAMP1 (H4A3) (cat # sc 200-11 1:400 IF), Galectin-3 (M3/38 sc-23938 1:800), 14-3-3 B-11 (sc-133232 1:1000) were from Santa Cruz; Flag M2 (cat. # F1804 1:1000 WB), ACTN2 EA-53 (# A7811 dilution 1:1000 IF) and actin AC-74 (# A2228 − 1:5000 WB) were from Sigma-Aldrich; HA.11 epitope tag (cat. 901513 − 1:1000) was from Biolegend; HA clone 3F10 (ref. 1186 7423001 1:800 FACS); Anti-HA High Affinity (ref.11867423001 1:500 IF) from Roche; Tomm20 clone 29 (cat # 612278 1:800 IF) and p62 Clone 3 (cat # 610832 1:800 WB) came from BD Biosciences. HRP-conjugated secondary antibodies for mouse (cat. # 401215 - dilution 1:5000) and rabbit (cat. # 401315 - dilution 1:5000) were from Calbiochem; donkey anti-rabbit IgG (H + L) Alexa Fluor 488 (cat. #A-21206 · dilution 1:500), Alexa Fluor 568 (cat. # A-10042 · dilution 1:500), donkey anti-mouse IgG (H + L) Alexa Fluor 568 (cat. # A-10037 − 1:500), Alexa Fluor 647 (cat. # A-31571 − 1:500), Alexa Fluor 594 (cat. # A-21203 − 1:500), donkey goat anti-goat IgG (H + L) Alexa Fluor 647 (cat. # A-21447 − 1:500), donkey anti-rat IgG (H + L) Alexa Fluor 647 (cat- #A21247 1:500) were from Thermo Fisher Scientific. The chemicals used were Torin 1 (cat. No. 4247) from Tocris; protease inhibitor cocktail (cat. no. P8340) and puromycin (cat. no. P9620) from Sigma-Aldrich; and PhosSTOP phosphatase inhibitor cocktail tablets (cat. no. 04906837001) from Roche.

### Clinical data collection
Clinical and laboratory data were collected from medical charts. Serum magnesium and potassium from patients affected by *RRAGD* mutations were compared with non-affected family members (pedigree numbers). Normality distribution was checked by using the Kolmogorov−Smirnov test and statistical analyses was carried out by two-sided unpaired t-test. Data collection and genetic analysis were performed after obtaining informed consent to publish identifiable medical information from all the subjects. The study protocol has been approved by the Ethical Committee of University of Campania "Luigi Vanvitelli" and is in compliance with all relevant ethical regulations.

### Whole-exome sequencing (WES)
Written informed consent was obtained from the parents. Genomic DNA was extracted from peripheral blood leukocytes using standard protocols. For library preparation of single samples, we followed the manufacturer's instructions (SureSelectQXT Automated Target Enrichment for the Illumina Platform, Protocol Version B0, November 2015, Agilent Technologies, Santa Clara, CA, USA). The WES of three affected members (X7321, X7322, X7323) was enriched using the SureSelect Human All Exon v7. Enriched DNA was validated and quantified by microfluidic analysis using the High Sensitivity D1000 ScreenTape Assay (Agilent Technologies) and the 4200 TapeStation System.

### Data analysis
The libraries were sequenced using the Illumina NovaSeq 6000 System performing paired-end runs covering $2 \times 150$ nt (Illumina Inc., San Diego, CA, USA). The generated sequences were analyzed using an in-house pipeline[47] designed to automate the analysis workflow, made up of modules performing each step using the appropriate tools commercially available or developed in-house. The average coverage of the target bases was 100x with 95.4% of the bases covered by at least 20 reads.

### Variant detection, mutation annotation, and Sanger sequencing of genomic DNA
Autosomal dominant inheritance mechanism was considered, attention was focused on variants that were present at a minor allele frequency of ≤0.001 in Genome Aggregation Database (gnomAD, https://gnomad.broadinstitute.org), (dbSNP, https://www.ncbi.nlm.nih.gov/snp/), Exome Aggregation Consortium (Exac, http://exac.broadinstitute.org), ClinVar database (https://www.ncbi.nlm.nih.gov/clinvar/) and in the internal database of ~5400 Italian subjects. All three probands showed a heterozygous missense variant in the *RRAGD* gene (NM_021244.5): c.263 C > T (p.Pro88Leu) on chromosome 6: 90097195 (hg.19). This variant was not previously reported in literature (LOVD, ClinVar, HGMD) and is absent in the frequency databases (gnomAD or ExAC). The identified candidate

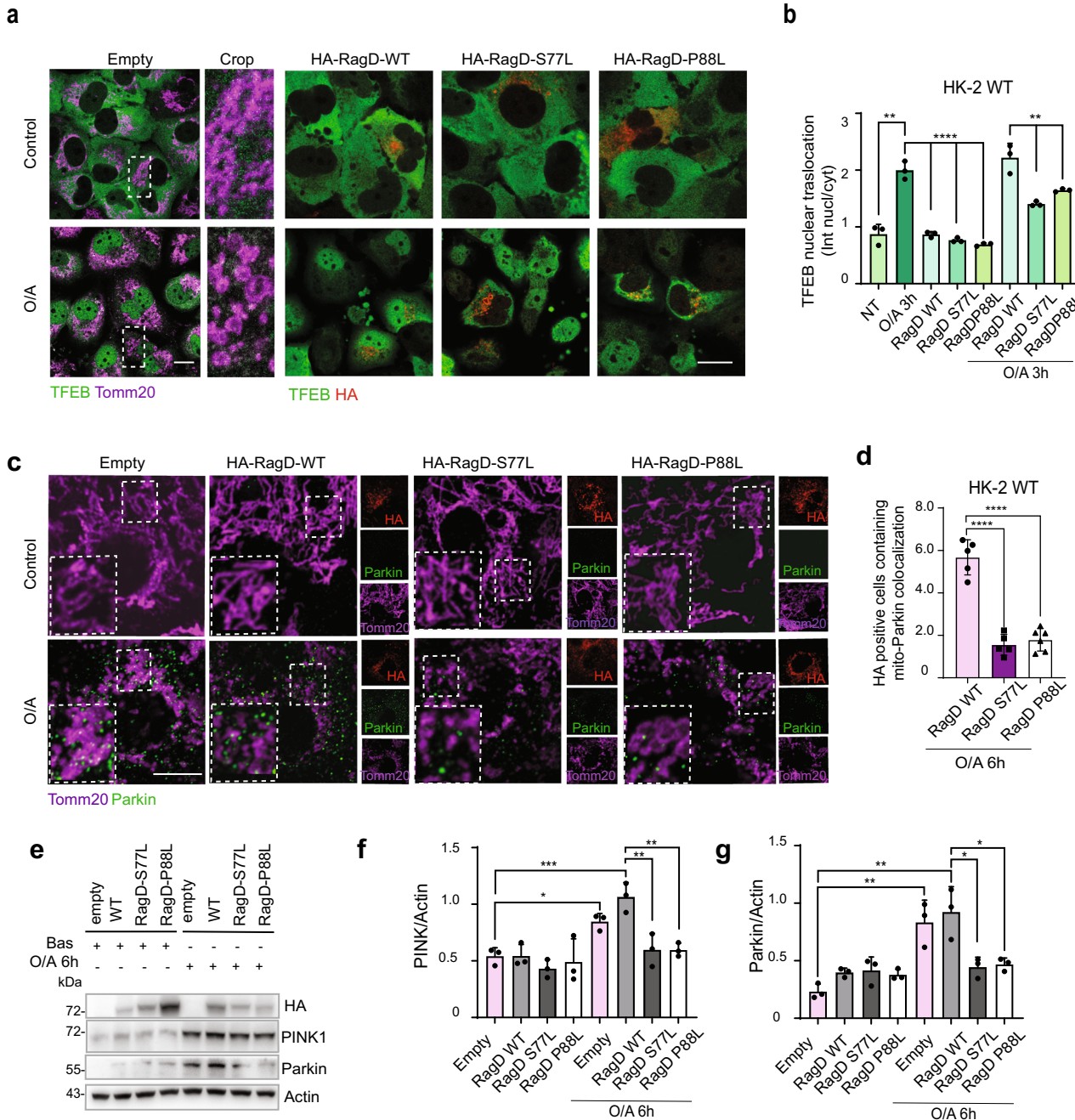

**Fig. 4 | RagD mutants impair TFEB response to mitochondrial injury.**
**a** Representative immunofluorescence images of HK-2 cells transfected with HA-RagD WT or mutants (S77L or P88L) and treated with 10 µg/ml Oligomycin and 5 µg/ml Antimycin A (O/A) for 3 h. Cells were stained with anti-TFEB and anti-HA. Staining for Tomm20 was used as mitochondrial marker. Scale bar, 10 µm. **b** Graph shows the TFEB nuclear-cytosolic ratio intensity of the HA positive (RagD WT or mutants S77L, P88L) cells treated with O/A (mean ± s.d. of *n* > 1000 cells from *n* = 3 independent experiments). Quantification performed as in Fig. 2f. Ordinary one-way ANOVA Tukey multiple comparison test (**\**p* < 0.01, **\*\*\****p* < 0.0001).
**c** Representative immunofluorescence images of HK-2 cells transfected with HA-RagD WT or mutants (S77L or P88L) and treated with 10 µg/ml Oligomycin and 5 µg/ml Antimycin A (O/A) for 6 h. Cells were stained with anti-Parkin (Rabbit

polyclonal) and anti-HA (Rat monoclonal). Staining for Tomm20 (Mouse monoclonal) was used as a mitochondrial marker. Scale bar, 10 µm. **d** Graph shows the mitochondria-Parkin co-localization of the HA positive (RagD WT or mutants S77L, P88L) cells treated with O/A (mean ± s.d. for *n* = 195 cells of 3 independent experiments). Ordinary one-way ANOVA Tukey multiple comparison test (**\*\*\****p* < 0.0001). **e** Representative western blot images showing: PINK1, Parkin, and Actin in HK-2 cells transiently transfected with HA-RagD-WT or RagD mutants, S77L, P88L, treated ± O/A for 6 h. Graphs show the PINK1/Actin (**f**) and Parkin/Actin ratios (**g**). Data are representative of *n* = 3 biologically independent experiments. Ordinary one-way ANOVA Tukey multiple comparison test (*\*p* < 0.05, **\**p* < 0.01, **\*\*\****p* < 0.001).

disease causing variant was confirmed in all affected members by direct Sanger Sequencing using specific primers (*RRAGD* exon 2: Forward 5′-TGGAAGAGTGGCATCATCTG-3′ Reverse 5′-CCATCA TACCCCTGAGACTC-3′). Healthy members did not show this variant.

## Protein stability and ligand binding prediction
We used the MERSi algorithm from Molsoft ICM-Pro 3.8-6a (Molsoft, San Diego, CA, USA) to predict the effect of the missense mutations in RagD on protein stability and protein nucleotide binding[48]. The effect of the individual mutations was analyzed using the PDB model of RagD

GTP with the reference code 2q3f. To calculate the stability change, the algorithm takes into consideration a comprehensive set of free-energy contributions, represented under the formula $\Delta/\Delta G = \Delta G_{mutant} - \Delta G_{WT}$. The residue-specific constants that account for the free energies of the different states were previously derived empirically using a large set of experimental data[48]. Each mutation was followed up by Monte Carlo simulations[49] for that particular residue and surrounding residues. Other areas of the protein are considered rigid for the purpose of this analysis. A positive energy value indicates that the mutation is likely to be destabilizing. Since no structural model is currently available for RagD GDP, we used the RagC GDP, PDB 6CES to estimate the effect of the mutations in the diphosphate-loaded form of the protein. The binding free energy change ($\Delta/\Delta G$ binding) was determined as the difference between the binding free energy of the mutant and wild-type proteins. As for the stability analysis, Monte Carlo simulations were performed to account for structure relaxation.

## Modeling the missense mutations in RagD
Structural models of the mutations in RagD were prepared using RagD GTP (PDB code 2q3f). Mutations were introduced using ICM-Pro 3.8-6a (Molsoft, San Diego, CA, USA) with local minimization to optimize side chain positions in the vicinity of the mutation site.

## Protein purification
Variants of recombinant GST-RagD (WT, S76L, P88L, T97P, P119R, and I221K) were expressed from pET24d(+) plasmid (Novagen) in *E. coli* Rosetta (DE3) cells (Novagen). Protein expression was induced with 1 mM IPTG overnight at 16 °C. The bacteria were disrupted by sonication in lysis buffer [50 mM Tris-Cl (pH 7.5), 200 mM NaCl, 5% glycerol, 1 mM DTT, 1x complete EDTA free protease inhibitor cocktail (Roche), and 0.1% Nonidet P40], and the clarified lysates were incubated at 4 °C for 1 h with Glutathione Sepharose 4B (GE Healthcare). The resin was then washed three times with lysis buffer and bound GST- RagD variants were eluted with 10 mM reduced glutathione, 50 mM Tris-Cl (pH 7.5), and 10% glycerol. The purified protein samples were analyzed by SDS-PAGE and Coomassie blue staining, and stored at −80 °C.

## GTP-binding assay
In GTP-binding assays[19], increasing concentrations (from 31.25 nM to 2 µM) of recombinant GST-RagD variants were incubated with 1 nM $\alpha$-$^{32}$P-GTP (Hartmann Analytics) for 4 h at 4 °C in 50 mM HEPES (pH 7.5), 100 mM potassium acetate, 2 mM $MgCl_2$, 2 mM DTT, and 0.1% CHAPS. The samples were then spotted onto a chilled metal block covered with Parafilm and to induce crosslinking, UV light of a total energy of 300 mJ was applied using GS Gene Linker (Bio-Rad). The samples were then mixed with Laemmli buffer, denatured for 10 min at 65 °C, and run on 7.5% SDS-PAGE gels that were then dried and analyzed by autoradiography.

## GTP hydrolysis assays
100 nM of purified GTPase were incubated for 3 h at room temperature in reaction buffer (20 mM Tris-HCl [pH 8.0], 2 mM EDTA, and 1 mM DTT, 20 mM $MgCl_2$) containing 40 nM [$\alpha$-$^{32}$P]-GTP (Hartman Analytic; 3,000 Ci/mmol). Reactions of 10 mL were stopped by the addition of 3 µL stop buffer (1% SDS, 25 mM EDTA, 5 mM GDP, and 5 µM GTP), samples were then denatured for 4 min at 65 °C, and 5 µL of each sample were loaded onto PEI Cellulose F Plates (Merck). Thin-layer chromatography was performed with buffer containing 1.0 M acetic acid and 0.8 M LiCl. Results were visualized using a phosphorimager and quantified with ImageJ software.

## Plasmids
pRK5-HA-GST-RagD wt (#19307) and pRK5-HA-GST-RagD S77L (#19308) were a kind gift from D. Sabatini (Addgene plasmids). The plasmids carrying the human mutations: pRK5- HA-GST-RagD **P88L**,

pRK5-HA-GST-RagD **S76L**, pRK5-HA-GST-RagD **S76W**, pRK5-HA-GST-RagD **P119L**, pRK5-HA-GST-RagD **P119R**, pRK5-HA-GST-RagD **I221K**, pRK5-HA-GST-RagD **T97P** were generated using the QuikChange II-E Site-Directed Mutagenesis Kit (no. 200555, Agilent Technologies). pTRIP-GPNMBprom-NUC-mCherry was generated by synthesis of the GPNMB promoter region sequence (448 bp) flanked by *NdeI* and *NheI* restriction enzyme sequences. The fragment was cloned into a lentiviral vector NUC-mCherry-pTRIP plasmid (was a gift from Thomas Weber Addgene cat. No. 163520) changing the CMV promoter sequence with the GPNMB promoter sequence between the *NdeI* and *NheI* restriction sites.

## Generation of HeLa *FLCN* KO and HK-2 *FLCN* KO cell lines
HeLa (ATCC CCL/2) *FLCN* knockout cells were generated using the all-in-one CRISPR/Cas9 vector system obtained from Merck. The gRNA sequence was 5′-TGTCAGCGATGTCAGCGAG-3′. A single plasmid containing the Cas9-GFP and the gRNA was transfected using the LipoLTX reagent (cat no. 94756 Thermofisher) following the manufacturer's instructions. 48 h after transfection, GFP-positive cells were sorted into single cells via FACS. Clones were analyzed by Sanger sequencing and the clones with homozygous indel mutations were expanded. HK-2 (ATCC CRL-2190) *FLCN* knockout cells were generated by CRISPr/Cas9 using the gRNA 5′-GGCACCATGAATGCCATCG-3′. Single-cell clones carrying the INDEL mutations were genotyped by PCR reaction with the specific primers: UP 5′-CTGCCTCCTCTGGTCATTCC-3′ and DOWN 5′-CTCGCACATGTCCGACTTTT-3′. PCR products were analyzed by DNA Sanger sequencing and the cell clone carrying the compound heterozygous deletion introducing a PSC: c.7 DEL CCATCGTG/c.17DELGCAGGCCTGTTGCAGTCTCCAAGGCACCATGAATGCCATCGT GGCTCTCTGCCAC was selected and expanded.

## Cell cultures
Cells were cultured in the following media: HeLa (ATCC CCL/2) and HeLa *FLCN* KO in MEM (cat. no. ECB2071L, Euroclone); HEK293T (CRL-3216) in DMEM high glucose (cat. no. ECM0728L, Euroclone); HK-2 (ATCC CRL-2190), HK-2 *FLCN* KO in DMEM-F12 (cat. no. 11320033, Thermo Fisher Scientific). All media were supplemented with 10% inactivated FBS (cat. no. ECS0180L, Euroclone), 2 mM glutamine (cat. no. ECB3000D, Euroclone), penicillin (100 IU/ml), and streptomycin (100 µg/ml) (cat. no. ECB3001D, Euroclone) and maintained at 37 °C and 5% $CO_2$. HeLa *FLCN* KO and HK-2 *FLCN* KO cells with inducible expression of TFEB-GFP were generated upon transduction of these cells with pLVX-TETONE-GFP-TFEB inducible lentiviral plasmids. HeLa *FLCN* KO GPNMBprom-NUC-mCherry were generated upon transduction of these cells with the pTRIP-GPNMBprom-NUC-mCherry plasmid. Cells were selected using 1 µg/ml of puromycin and then FACS sorted. All cell lines were purchased from ATCC, validated by morphological analysis, and routinely tested for the absence of mycoplasma. Fibroblasts from *RRAGD* P88L patients and healthy subjects were cultured in DMEM high glucose supplemented with 20% inactivated FBS, 2 mM glutamine, penicillin (100 IU/ml), and streptomycin (100 µg/ml) and maintained at 37 °C and 5% $CO_2$.

## Culture of human pluripotent stems and in vitro differentiation into cardiomyocytes
Human induced pluripotent stem cells (hiPSCs) LUMC0020iCTRL-06 line registered at (https://hpscreg.eu/cell-line/LUMCi028-A and derived from skin fibroblasts of a healthy female individual after informed consent and following approval from the Leiden University Medical Center committee[50]) were cultured as previously described[51,52]. Briefly, hiPSCs were seeded on vitronectin recombinant human protein and cultured in E8 medium; cells were passaged twice a week using PBS containing EDTA 0.5 mM. RevitaCell Supplement (1:200) was added during passaging (all from Thermo Fisher Scientific).

Differentiation into cardiomyocytes (hiPSC-CMs) was induced in monolayer as previously described[51,52]. Briefly, $2.5 \times 10^4$ cells per cm² were seeded on growth factor-reduced Matrigel (Corning)-coated plates. 24 h after plating, cardiac mesoderm induction was established by culturing the cells in LI-BPEL medium supplemented with a mixture of cytokines (20 ng/mL BMP4, R&D; 20 ng/mL Activin A, Miltenyi Biotec) and the GSK3 inhibitor CHIR99021 (1.5 μmol/L, Axon Medchem). After 72 h, cytokines were removed and the Wnt inhibitor XAV939 (5 mM, Tocris) was added for 72 h. LI-BPEL medium was further refreshed every 3–4 days.

## Cell culture treatments

For drug treatment experiments, cells were incubated in a medium containing one or more of the following compounds: LLOMe methyl ester hydrobromide was purchased from Santa Cruz (Cat no. sc-285992). Cells were treated with 500 μM LLOMe for 1 h or 6 h and collected for immunofluorescence and WB. MK6-83 was purchased from Tocris Bioscience (cat. no. 5547). Cells were treated with 30 μM MK6-83 for 1 h or 6 h and collected for immunofluorescence and WB. Cells were treated with a mixture of 10 μg/ml Oligomycin (cat no. CAS 1404-19-9 EMD Millipore) and 5 μg/ml Antimycin A (cat no. 1397-94-0 Sigma-Aldrich) (O/A) for 3 h or 6 h and collected for immunofluorescence and WB, respectively. For experiments involving amino acid starvation, cells were rinsed twice with PBS and incubated for 1 h (6 h only for hiPSC-CMs experiments) in amino acid-free RPMI (cat. no. R9010-01, US Biological) supplemented with 10% dialyzed FBS. For amino acid refeeding, cells were re-stimulated for 45 min with 1× water-solubilized mix of essential (cat. no. 11130036, Thermo Fisher Scientific) and non-essential (cat. no. 11140035, Thermo Fisher Scientific) amino acids resuspended in amino-acid-free RPMI supplemented with 10% dialyzed FBS, plus glutamine.

## Western blot

Cells were rinsed once with PBS and lysed in ice-cold lysis buffer (250 mM NaCl, 1% Triton, 25 mM Hepes pH 7.4). Directly before use, protease/phosphatase inhibitors (Thermo Fisher Scientific) were added to the lysis buffer. Total lysates were passed 10 times through a 25-gauge needle with syringe, kept at 4 °C for 10 min and then cleared by centrifugation in a microcentrifuge ($18,800 \times g$ at 4 °C for 10 min). Protein concentration was measured by BCA assay (Cat no. 23225 Thermo Scientific). Cell lysates were resolved on SDS-polyacrylamide gel electrophoresis on 4–12% Bis-Tris gradient gels (cat. no. MP41G15 mPage Bis-Tris Precast gels, Millipore) then transferred to PVDF membranes (Millipore Corp ref IPVH00010). Membranes were incubated with primary antibodies overnight at 4 °C and with secondary antibodies for 1 h at RT. Quantification of western blotting was performed by calculating the intensity of phosphorylated and total proteins by densitometry analysis using the Fiji software. The ratios between phosphorylated and total protein values were normalized to a control condition for each experiment.

## Immunoprecipitation

HeLa *FLCN* KO cells were transfected with TFEB-FLAG in combination with RagD-HA-GST WT and mutants and RagA-HA-GST. 48 h following transfection, cells were rinsed twice with ice-cold PBS and lysed with NP-40 lysis buffer (50 mM Tris-HCl pH 7.5, 100 mM NaCl, 0.5% NP-40, one tablet of phosphatase inhibitor cocktail (PhosphoSTOP cat. no. 04906837001) per 10 ml of buffer and protease inhibitor (P8340 SIGMA). The lysates were cleared by centrifugation at $18,800 \times g$ at 4 °C in a microcentrifuge for 10 min. Protein concentration was analyzed by Bradford assay (BIORAD Cat no #5000006). For immunoprecipitation, the FLAG M2 (A2220 Sigma) beads were pre-equilibrated in NP-40 lysis buffer and then added to cleared lysates and incubated at 4 °C for 2 h. After immunoprecipitation, the beads were washed three times with wash buffer (50 mM Tris-HCl pH 7.5, 0.2% NP-40,

50 mM NaCl) supplemented with protease and phosphatase inhibitors. Immunoprecipitated proteins were denatured by the addition of Laemmli buffer, heated at 95 °C for 10 min, and centrifuged at max speed for 5 min. Immunoprecipitated and input samples were resolved by SDS-polyacrylamide gel electrophoresis on 4–12% Bis-Tris gradient gels (cat. no. MP41G15 mPage Bis-Tris Precast gels, Millipore) and analyzed by immunoblotting with the indicated primary antibodies.

For endogenous IP, HeLa *FLCN* KO cells were transfected with RagD-HA-GST WT and mutants in combination with RagA-HA-GST. 48 h following transfection cells were crosslinked using DSP (Cat. no 22585 Thermofisher Scientific) and lysed with RIPA lysis buffer (HEPES 40 mM pH 7.4, EDTA 2 mM, 1% Na-Deoxycholate, 1% NP-40, 0.1% SDS, 10 mM Na₄-Pyrophosphate, 10 mM Na₂-Glycerophosphate, 1X protease inhibitors). The lysates were cleared by centrifugation at $18,800 \times g$ at 4 °C in a microcentrifuge for 10 min. Protein concentration was analyzed by Bradford assay. For immunoprecipitation, the HA beads (Thermofisher Scientific ref 26182) were pre-equilibrated in RIPA lysis buffer and then added to cleared lysates and incubated at 4 °C overnight. After immunoprecipitation, the beads were washed three times with RIPA buffer supplemented with 250 mM NaCl and immunoprecipitated proteins were denatured and resolved by Western blot.

## FACS processing and GPNMB fluorescent reporter activity

To measure the *GPNMB* promoter activity, HeLa *FLCN* KO cells carrying the GPNMBprom-NUC-mCherry were transfected with appropriate RagD constructs for 48 h and then processed for FACS analysis. Cells were trypsinized and fixed in Fix/Perm Buffer (cat no. 51-9008100 BD Biosciences) for 10 min at 37 °C. Cells were pelleted at $0.4 \times g$ for 5 min and washed with Perm/Wash buffer (cat no. 51-9008102 BD Biosciences). Then, cells were permeabilized with Perm/Wash buffer for 30 min on ice and then for 30 min at RT. After permeabilization, cells were incubated with anti-HA (clone 3F10 ref. 11867423001 Roche 1:800 FACS) primary antibody overnight at 4 °C to isolate the HA-RagD positive cells. Upon staining, cells were washed three times and incubated with anti-rat secondary antibody (Alexa Fluor 488 cat no. A11006 1:500 Thermo Fisher Scientific) for 1 h at RT. Then cells were washed three times and resuspended in FACS buffer (2 mM EDTA, 2% FBS diluted in PBS). The fluorescent intensities were analyzed by FACS (FACS ARIA III BD biosciences).

## RNA extraction, reverse transcription, and quantitative PCR

RNA samples from cells were obtained using the RNeasy kit (Cat no. 74134 Qiagen). according to the manufacturer's instructions. cDNA was synthesized using QuantiTect Reverse Transcription kit (Cat no. 205314 Qiagen). Real-time quantitative RT-PCR on cDNAs was carried out with the LightCycler 480 SYBR Green I mix (Cat no. 04887352001 Roche) using the Light Cycler 480 II detection system (Roche) with the following conditions: 95 °C, 5 min (95 °C, 10 s; 60 °C, 10 s; 72 °C, 15 s) × 40. Fold change values were calculated using the ΔΔCt method. In brief, an internal control (*HPRT1*) was used as a 'normalizer' to calculate the ΔCt value. Next, the ΔΔCt value was calculated between the 'control' group and the 'experimental' group. Finally, the fold change was calculated using 2(−ΔΔCt). Biological replicates were grouped in the calculation of the fold-change values. For TFEB targets gene analysis HeLa *FLCN* KO cells were co-transfected with either RagD mutants constructs and EGFP empty vector using the Fugene reagent (cat no. E2312 Promega) following the manufacturer's instructions. 48 h upon transfection, GFP-positive cells were sorted via FACS and RNA was extracted for the subsequent analysis.

## Immunofluorescence and image quantification

For immunofluorescence, the following antibodies were used: TFEB (Cell Signaling cat. 4240S 1:200), Phospho-TFEB (Ser211) (Cell Signaling cat. 37681 1:200), HA.11 clone 16B12 (BioLegend 901501 1:500) Parkin (Cell Signaling cat 2132 1:200), Tomm20 (BD cat n 612278

 

1:800), Galectin-3 (M3/38 sc-23938 1:800) already described above. Cells were fixed in PFA 4% for 10 min and permeabilized with saponin blocking buffer whereas for TFEB immunostaining cells were permeabilized in 0.1% (w/v) Triton X-100, 1% (w/v) horse serum, and 1% (w/v) BSA in PBS. Cells were incubated with the indicated primary antibodies overnight and subsequently incubated with secondary antibodies for 1 h (AlexaFluor 488, AlexaFluor 568, AlexaFluor 647 all Thermo Fisher 1:400). For confocal imaging, the samples were examined under a Zeiss LSM 800 confocal microscope. Optical sections were obtained under a ×63 immersion objective at a definition of 1024 × 1024 pixels, adjusting the pinhole diameter to 1 Airy unit for each emission channel to have all the intensity values between 1 and 254 (linear range). The microscope was operated on the Zeiss Zen blue 2.1 software platform (Carl Zeiss). For image analysis, the images were acquired with an automated confocal microscopy (Opera System, Perkin-Elmer) and analyzed through Columbus Image Data Storage and Analysis System (Perkin-Elmer, Waltham, MA, USA) by the High Content Screen (HCS) Facility at Tigem (Opera Phenix HCS system, Perkin-Elmer). A dedicated script was applied, as previously reported[53], to evaluate TFEB nuclear translocation and mTORC1/Lamp1 co-localization. At least three independent experiments, and up to 3000 individual cells per treatment from at least two independent wells, were routinely analyzed. P values were calculated on the basis of mean values from independent wells. The data are represented by the percentage of nuclear translocation versus the indicated control using Excel (Microsoft) or Prism software (GraphPad software).

### In vitro generation of RagD mRNA and transfection

RagD-WT and RagD mutants (S76L, P88L, T97P) sequences were ordered as a gBlock (IDT), including the T7 promoter, HA-tag, target sequence, T2A self-cleaving peptide sequence and mCherry, and cloned into the pMiniT 2.0 vector using the NEB PCR Cloning Kit (New England Biolabs) according to the manufacturer's instructions. The target gBlock sequences were amplified by PCR using Platinum Taq DNA polymerase (Thermofisher Scientific) and the primers provided by the NEB PCR Cloning Kit (New England Biolabs). The PCR products were purified and subsequently digested with *XhoI* restriction enzyme and run on a 1% agarose TBE gel. The specific band was extracted using the Wizard SV Gel and PCR Clean-Up System (Promega) and used as a template for in vitro transcription.

RagD-WT and RagD mutants (S76L, P88L, T97P) mRNAs were generated by in vitro transcription using INCOGNITO T7 5mC- and Ψ-RNA Transcription Kit, ScriptCap Cap 1 Capping System and A-Plus Poly(A) Polymerase Tailing Kit (all from Cellscript, LLC) following the manufacturer's instructions. hiPSC-CMs were transfected with RagD-WT or RagD mutants mRNAs using Lipofectamine Stem Transfection Reagent (Invitrogen) according to the manufacturer's instructions. For transfection, 250 ng of the mRNA per 50,000 cells was mixed with Opti-MEM (Gibco) and 0.5 µl Lipofectamine Stem Transfection Reagent in a total volume of 15 µl and incubated for 10 min at RT before being added drop-wise to the cells. The cells were refreshed 18–20 h after transfection. The day after, hiPSC- CMs were treated with 500 µM LLOMe for 1 h and collected for immunofluorescence.

### Immunofluorescence analysis of hiPSC-CMs

HiPSC-CMs were fixed with 4% PFA in PBS for 10 min and permeabilised with PBS/0.1% Triton-X-100 for 7 min at room temperature. The cells were then blocked in blocking solution containing PBS/1% BSA for 1 h at room temperature and incubated overnight at 4 °C with anti-alpha sarcomeric actinin (ACTN2 mouse monoclonal, Sigma-Aldrich #A7811, dilution 1:1000) and anti-TFEB (rabbit polyclonal, Cell Signaling #4240S, dilution 1:200). After washing, hiPSC-CMs were incubated for 1 h at room temperature in the dark with the appropriate fluorochrome-conjugated secondary antibody Donkey anti-Rabbit Alexa Fluor 488 (1:300) and Donkey-anti-mouse AF594 (1:300).

Nuclei were counterstained with DAPI (all Thermo Fisher Scientific). Images were acquired with a Leica SP8 WLL confocal laser-scanning microscope using ×63 magnification objective and Z stack acquisition.

### Mammalian lentiviral production and hiPSC-CMs transduction

Lentiviruses were produced by transfection of HEK293T cells with pLVX-EIF1α-HA-GST- RagD-WT, pLVX-EIF1α-HA-GST-RagD-P88L and pLVX-EIF1α-HA-GST-RagD-S76L, constructs in combination with the psPAX2 [Addgene, #12260], pMD2G/VSVG [Addgene, #12259] packaging plasmids using Lipofectamine LTX transfection reagent (Invitrogen). Five hours after transfection, the medium was changed to DMEM supplemented with 10% FBS. Forty-eight hours later, virus-containing supernatants were collected, passed through a 0.45-µm filter to eliminate cell debris. hiPSCs (CBiPS1sv-4F-40, derived from cord blood of a neonatal female individual, registered at https://hpscreg.eu/cell-line/ESi007-A and supplied by EBiSC-European bank for induced pluripotent stem cells https://ebisc.org/ESi007-A) were cultured on human recombinant truncated vitronectin protein and cultured in E8 medium; cells were passaged twice a week with PBS containing EDTA 0.5 mM.

Cells were then differentiated into cardiomyocytes using the STEMdiff Cardiomyocyte Differentiation Kit (Stem Cell Technologies, #05010) and subjected to lentiviral infection (Multiplicity Of Infection: 3) at Day 13 (beating appeared at day 8) of differentiation in the presence of 8 µg/ml polybrene (cat. no. tr-1003-G, EMD Millipore).

### Statistical analysis

The experiments were repeated at least three times, unless stated otherwise. As indicated in the figure legends, all quantitative data are presented as the mean ± s.d. or SEM of biologically independent experiments or samples. For each experiment we described specific statistic test used and the relative significance in the figure legend. Statistical analyses were performed using GraphPad Prism 8.0. For Real Time experiments, a script for linear regression was designed and used in R software for qpcR (version 1.4-1).

### Reporting summary

Further information on research design is available in the Nature Portfolio Reporting Summary linked to this article.

## Data availability

Full scans for all western blots as well as raw data for all the graphs are provided with this manuscript as Supplementary Fig. 6 and Source data file, respectively. For graphs, the exact *p* value for all the experiments is present in the Source data file. The Whole-Exome Sequencing (WES) data are deposited in Sequence Read Archive (SRA) of NCBI repository (BioProject ID: PRJNA960632, available at the following link). All other data are available from the corresponding author on request. Source data are provided with this paper.

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

## Acknowledgements

We are grateful to Drs. Gennaro Napolitano, Chiara Di Malta, Graciana Diez-Roux, Carmine Settembre for helpful suggestions and for the critical reading of the manuscript. We also thank Malika Jaquenoud for technical assistance. We also want to thank the High Content Screening (HCS) Facility of Tigem and Lucia Perone from Cell Culturing Core of Tigem for technical assistance. We thank Rosaria Tuzzi for technical assistance during isolation of Pro88Leu patient-derived fibroblasts. We thank Eugenio del Prete from the Tigem Bioinformatic Core, for the statistical analysis of Real Time PCR. We thank Luigi Ferrante for technical assistance during FACS analysis. We thank Dorien Ward-van Oostwaard (Leiden University Medical Center, LUMC) for technical support in hiPSC-CM differentiation and Sanne Wiersma (LUMC) for help with mRNA in vitro transcription. We thank Cathal Wilson for the English proofreading of the manuscript. This work was supported by the Italian Telethon Foundation, European Research Council H2020 AdG (LYSO-SOMICS 694282 to A.B.) Associazione Italiana per la Ricerca sul Cancro A.I.R.C. (IG-22103 to A.B.), MIUR (PRIN 2017E5L5P3 and PRIN 202032AZT3 to A.B.), the Swiss National Science Foundation (310030_184671 to C.D.V.), European Research Council (ERC-CoG 101001746 Mini-HEART to M.B.), and the Italian Telethon Foundation (TGM22CBDM11 to L.S.).

## Author contributions

I.S., M.F., and A.B. conceived the study. I.S. designed, performed, and analyzed most of the experiments of immunofluorescences, WB, RT-qPCR. M.F. performed the co-immunoprecipitations and generated and characterized the GPNMBprom-mCherry reporter. N.Z. performed RagD plasmids mutagenesis and virus and provided technical support to the experiments. C.V. was involved in some immunofluorescence on FLCN-knockout cells and generated Hela FLCN KO CRISPR–Cas9 gene-edited cell line. L.D. and C.D.V. performed and analyzed the GTP-binding and GTP-hydrolysis assay. F.D.V.B, A.T., and V.N. performed and analyzed the WES and Sanger DNA sequencing from patients. J.M. generated HK-2 FLCN KO CRISPR–Cas9 gene-edited cell line and performed some microscopy and FACS experiments. G.C. generated the RagD WT and RagD mutants mRNAs. G.C. and V.M. performed and analyzed the experiments on hiPSC-CMs. M.B. supervised hiPSC experiments. M.T. and L.S. performed and analyzed the experiments on lentiviral infection of iPSC-CMs. G.P. and S.F. performed isolation of Pro88Leu patient-derived fibroblasts. F.T. and C.R. performed patients data collection and family pedigree. M.E.G.d.A. and L.A.H. performed and analyzed the in silico modeling of RagD protein structure. I.S., M.F., and A.B. wrote the manuscript. A.B. supervised the study.

## Competing interests

A.B. is cofounder of CASMA Therapeutics, Inc, and Advisory board member of Avilar andCoave Therapeutics and of Next Generation Diagnostics. The remaining authors declare no competing interests.

## Additional information

[1]Telethon Institute of Genetics and Medicine (TIGEM), Pozzuoli, (NA), Italy. [2]Medical Genetics Unit, Department of Medical and Translational Science, Federico II University, Naples, Italy. [3]Leiden University Medical Center, 2333ZC Leiden, the Netherlands. [4]Institute of Cell Biology, Biocenter, Medical University of Innsbruck, Innsbruck, Austria. [5]Department of Biology, University of Fribourg, CH-1700 Fribourg, Switzerland. [6]Department of Precision Medicine, University of Campania "Luigi Vanvitelli", Naples, Italy. [7]Department of Translational Medical Sciences, University of Campania "L. Vanvitelli", Naples, Italy. [8]Institute for Genetic and Biomedical Research, National Research Council (CNR), Milan, Italy. [9]Department of Biology, University of Padua, 35131 Padua, Italy. [10]Veneto Institute of Molecular Medicine, 35129 Padua, Italy. [11]Biogem Research Institute Ariano Irpino, Ariano Irpino, Italy. [12]Department of Molecular and Human Genetics, Baylor College of Medicine, Houston, TX, USA. [13]Jan and Dan Duncan Neurological Research Institute, Texas Children's Hospital, Houston, TX, USA. [14]These authors contributed equally: Irene Sambri, Marco Ferniani. ✉e-mail: ballabio@tigem.it

