## [Peer Review File · Nature Communications]

RagD auto-activating mutations impair MiT/TFE activity in kidney tubulopathy and cardiomyopathy syndromeReviewers' comments:

Reviewer #1 (Remarks to the Author):

In this manuscript, the authors showed that RagD mutations activating mTORC1 in human patients with kidney tubulopathy and cardiomyopathy syndromes were identified and that these RagD mutations affected MiT/TFE activity. However, another research group has already reported overlapping RagD mutations in patients with kidney tubulopathy and cardiomyopathy syndromes and the results displayed by the authors are not sufficient to show the role of MiT/TFE. Unfortunately, this paper seems to be less clear that the authors provide a scientific advance for publication in Nature Communications.

Major concerns,

1. In Fig. 1E, the authors showed the different roles of RagD mutations on RagD protein. However, there were no supporting evidences in the manuscript.
2. In Fig. 1F, GTP-bound levels of RagD mutants were shown using GTP- conjugated bead. The authors also should check GTP status of RagD mutants by enzymatic GTPase assay method.
3. In Fig. 2, Fig. 3, Supplementary Fig.1, and Supplementary Fig.2, the authors should also check the results in control cells (FLCN WT). Furthermore, the author only checked the effect of RagD mutations in FLCN KO cells. Leucyl-tRNA synthetase 1 has been known to be a RagD-GAP and the authors should also check the effects of RagD mutations in leucyl-tRNA synthetase 1 KO cells.
4. In Fig. 2A, the authors showed that exogenous FLAG-tagged TFEB coimmunoprecipitated with RagA/RagD heterodimer containing RagD mutations. However, this result does not reveal whether TFEB binds to RagA or RagD. The authors should check whether endogenous TFEB also binds to RagA or RagD and that this co-immunoprecipitation is mTORC1 dependent. In addition, RagD has been known to form a complex with RagB rather than RagA. What happens to TFEB binding if RagB is overexpressed instead of RagA?
5. In Supplementary Fig. 1A, the authors showed that RagD mutations induced TFEB phosphorylation. However, the data on the mechanisms by which RagD mutations induce TFEB phosphorylation are lacking.
6. In Fig. 2B, if FLCN is a GAP for RagC/D, p-S6K level should be reduced in FLCN KO cells than in WT cells. But, we cannot confirm this. Also, since there is no effect of RagD mutations on p-S6K, the results of mTORC1-activating RagD mutations cannot be confirmed.
7. In Fig. 2 and Supplementary Fig.1, the authors used FLCN KO cells. The authors should check that these cells are actually FLCN KO.
8. In Supplementary fig. 1C, the authors showed lysosomal localization of RagD WT, S77L and P88L. What about other RagD mutants? The authors should quantify the result of Supplementary Fig. 1C.
9. In Supplementary Fig. 1D and E, the authors should confirm whether TFEB is also localized in the lysosomes with RagD through Immunofluorescence or lysosomal fraction.
10. The suppression of nuclear localization of TFEB by exogenous RagD mutants may be an artificial effect. It should be checked whether the cellular localization of TFEB is regulated through the introduction of endogenous RagD mutations.
11. In Fig. 3A and Supplementary Fig. 2, the author confirmed changes in TFEB nuclear translocation by LLOMe, MK6-83 and O/A treatment. So what about autophagy system? Will it recover normally?
12. In Fig. 3, all experiments were done in control cells (FLCN WT). What about FLCN KO cells?
13. In Fig. 3A, the authors expressed statistical significance between control-transfected, LLOMe-treated group and RagD WT-transfected, LLOMe-treated group. However, there appears to be no statistical significance between the two groups.
14. In this manuscript, there are no supportive data on the functional importance of TFEB regulation by RagD mutations in kidney tubulopathy and cardiomyopathy syndromes.

Minor concerns,

1. In supplementary Fig. 1, the author overexpressed FLAG-TFEB, but what is GFP-TFEB?
2. In HA-RagD blot of Fig. 3D, overexpression of RagD S77L does not appear to increase RagD expression correctly.

Reviewer #2 (Remarks to the Author):

In this manuscript, the authors have conducted a very interesting study showing that disease-causing mutations in kidney tubulopathy and cardiomyopathy could lead to an auto-activation of RagD. These pathogenic mutants induced constitutive phosphorylation of TFEB by non-canonical mTORC1 signaling and inhibited the nuclear translocation and transcriptional activity of TFEB in cells lacking FLCN, or in cells under lysosomal damage/mitochondrial stress. In conclusion, the authors claim that inhibition of MiT-TFE transcription factors such as TFEB, but not hyperactivation mTORC1 signaling, contributes to the pathogenesis of kidney tubulopathy and cardiomyopathy.

Although there are some issues need to be addressed, this manuscript is overall promising and will be of sufficient interest to the Nat Commun readers. The manuscript contains logical experimental design and convincing cell biology and biochemical studies. The authors put effort to show the pathogenic role of RagD activation in TFEB-related disease. It will not only help to understand the basic cell biological mechanism, but also shed light to the potential therapeutic applications bases on the findings in this paper.

Specific comments on the manuscript are as follows:

1. The authors claimed that "Co-immunoprecipitation (Co-IP) analysis between RagD and TFEB showed that all RagD disease-causing mutations rescued the ability of RagD to interact with TFEB... (Figure 2A)". However, the protein levels of FLAG-TFEB in the (1%) input blot were different (e.g., S76L and T97P levels were much higher than WT). Therefore, it is recommended to re-perform the Co-IP assay using similar amount of FLAG-TFEB WT and mutants.
2. There is no formal demonstration of the "lysosomal" association between RagD and TFEB – if RagD physically interacts with TFEB and thereby allows mTORC1 to phosphorylate TFEB, one may speculate that TFEB (and active RagD) can colocalize with lysosome (Lamp1).
3. The inhibition of the expression of TFEB-targeting genes in Fig 2G is convincing. However, to fully support that the inhibition relies on the regulation of TFEB, these experiments should be also done in TFEB knockdown/knockout cells (or TFEB/TFE3/MITF triple KD/KO cells).
4. Fig 3D: the reduction of both PINK1 and Parkin protein levels is interesting. In fact the "activation" of PINK1/Parkin-mediated mitophagy does not necessarily rely on the protein level of Parkin, but requires Parkin mitochondrial translocation. Therefore, cellular localization of Parkin should be carefully examined to show mitophagy activity. In this case expression of exogenous Parkin (e.g., Parkin with a fluorescent tag or an epitope tag) could be helpful.
5. The reviewer suggests the authors to move at least some of the Supplementary Figures to the main figures. In addition, pTFEB (S211) immunoblots need to be shown in Supplementary Fig 1B, and the quality of Supplementary Fig 2C (lower panels) need to be improved.
6. The manuscript is overall well written, but multiple errors appear in the text and figures (e.g., Supplementary Fig 1A: if the labelings were correct, HA-RagA and FLAG-TFEB (GFP-TFEB?) were supposed to be detected on the left panels.

Reviewer #3 (Remarks to the Author):

"RagD autoactivating mutations impair MiT/TFE activity leading to kidney tubulopathy and cardiomyopathy" describes a novel RRAGD patient mutation, p.Pro88Leu, associated with kidney tubulopathy and cardiomyopathy. This is the second report of RRAGD-related kidney tubulopathy and cardiomyopathy syndrome, following a recent publication of Schlingmann et al.. The authors examined the structural consequence of the current known RagD mutations and further looked into the GTP-binding capability of these mutants that resulted in impaired GTP binding, demonstrating a gain of functionality. Furthermore, the authors, consistent with a previous publication by Kim et

al., demonstrated that this new RagD mutation along with previously reported mutations led to impaired regulation of the MiT-TFE factors and consequent dysregulation of lysosomal/autophagic response.

Overall, this work presents interesting findings that will contribute to more understanding of the Rag GTPases functionality. However, the molecular pathogenesis of the kidney and heart disease caused by RagD mutations remains elusive and there are several contradictions with previous literature.

Please find my comments below:

Major comments

1. The main issue of the manuscript is that the authors only provide cellular data in HeLa and HK-2 cells to study the role of RRAGD mutations. Although these experimental results are novel and interesting, they do not support claims about the molecular causes of cardiomyopathy and tubulopathy. As both mTOR hyperactivation and TFEB loss-of-function are associated with cardiomyopathy. Additional data from in vivo models would be required to answer this question. Although I understand that this may be beyond the feasibility of this revision, claims about the role of TFEB/mitophagy in the pathogenesis of the disease in the abstract and discussion section should be prevented.

2. The authors showed that mitophagy is indeed impaired in RagD mutants-expressing cells through reduction of Parkin and PINK1 proteins after treatment with ATP synthase inhibitor oligomycin and the complex III inhibitor antimycin A (O/A). Moreover, TFEB is retained in the cytoplasm in these cells upon O/A treatment. However, there is a lack of experiments to further demonstrate that specifically mitophagy is responsible for the kidney tubulopathy and cardiomyopathy in the patients. Are biopsies or patient cells (fibroblasts?) available that could confirm mitophagy in patients?

3. Previous studies have identified downstream targets of TFEB in cardiac dysfunction and mitochondrial signaling. Have downstream targets like GATA4 (Song et al.) or MCU and calcineurin signaling (Kenny et al.) been examined?

3. The authors do not demonstrate effects of previously published RRAGD mutations on p70S6K phosphorylation. This is in contrast with previously published data by Schlingmann et al. However, the experimental conditions are slightly different (no IP). What is the percentage of RagD-transfected HeLa FLCN KO cells? Could the phosphorylation be mostly dependent on non-transfected cells? Could low transfection efficiency in these cells be the reason that there were no significant changes in phosphorylation of 70S6K?

4. Along the same lines, the data demonstrating reduced lysosomal localization (Suppl Fig 1C) is in contradiction with the paper of Schlingmann et al., demonstrating increased interaction of a RagD mutant with mTOR and lysosomal V-ATPases. Could this be explained by the differences in cell type?

5. What are the authors' conclusions regarding the role of mTORC1 phosphorylation in TFEB functionality? Figure 2B and supplementary figure 1B nicely demonstrated that mTORC1 non-canonical signalling of TFEB phosphorylation is increased when RagD mutants are present in the cells, but did not elaborate further on this. Moreover, the results on LLOME and MK6-83 treatments seem to contradict the concept of mTORC1 phosphorylation-dependent as nuclear translocation is still not induced given that mTORC1-mediated TFEB phosphorylation was inhibited in these cells.

6. In the discussion section, the authors discuss quite thoroughly the roles of mitochondria in kidney and heart and how TFEB contributes to mitochondria quality control. While this hypothesis is promising, discussion on other results shown in the manuscript is missing. The discussion section now mainly cites supportive results, it would be important to also highlight the contradictions with the literature. E.g. Kim et al. demonstrate that mTOR inhibition did not ameliorate the cardiac phenotype in zebrafish, whereas this manuscript claims that mTOR

inhibition decreases TFEB phosphorylation. E.g. the above mentioned issues with the Schlingmann paper.

7. Interestingly, the authors describe the first RRAGD mutation outside of the G-boxes, could they speculate on the role of the G-boxes for the disease?

Minor comments

1. Please check throughout the manuscript the correct use of gene names RRAGD/RRAGC (italics) and protein names RagC/RagD.
2. Figure 2B
 - a. What could be the reason that mutations on the S76 and S77 sites resulted in higher TFEB levels compared to the other mutations?
 - b. In Figure 2B and Supplementary fig 1B, molecular weight shift in TFEB is considered as change in phosphorylation. However, pTFEB (S211) antibody was used in Supplementary Figure 1A. Could the authors comment on the different detection methods between these?
3. Figure 2E
 - a. Typo in the legend (representative should be representative)
 - b. What is the basal fluorescent of the reporter in non-transfected cells?
4. Figure 2F
 - a. What does "% of PE fluorescent" exactly represent? Intensity of the fluorophore? Was this normalised to anything?
5. Figure 3A, B, and C:
 - a. Images shown should be representative of the quantification graphs shown rather than effects of drug treatments in WT cells
6. Figure 3D:
 - a. It is not really clear what is quantified here. Please add the y-axis titles to the graphs
7. Supplementary Figure 1A:
 - a. Typo mutation name S211K should be I221K
 - b. In the figure, FLAG-TFEB was used for transfection but GFP-TFEB was used to label the blot. Is this correct?
 - c. In RagD empty condition, TFEB and RagA are also absent. Was this perhaps a triple empty condition?

Reviewer #4 (Remarks to the Author):

Irene Sambri et al described that a novel mutation in RRAGD can lead to kidney tubulopathy and cardiomyopathy via inhibiting MiT-TFE factors but not mTORC1 hyperactivation. The study is interesting but is insufficient for NC at present.

major issues:

1. All the mutations studies was conducted in Hela cells with plasmids transfection, which can not prove the viewpoint proposed by the authors. I recommend the authors to construct the mutant kidney cell line and cardiomyocytes by Cas9.
2. The study need more evidence to prove the viewpoint in animal models such as point mutation knock in mice.
3. The genetic relationship between RRAGD and kidney tubulopathy and cardiomyopathy have been reported in J Am Soc Nephrol . 2021 Nov;32(11):2885-2899. I recommend to detect more other novel mutations.

mirror issues:

1. the pedigree of the family may have some problems, such as II-3, II-4, II-5 and II-6.
2. several data were showed in mRNA levels which is not enough, please provide the WB data.

Reviewer #1 (Remarks to the Author):

In this manuscript, the authors showed that RagD mutations activating mTORC1 in human patients with kidney tubulopathy and cardiomyopathy syndromes were identified and that these RagD mutations affected MiT/TFE activity.

However, another research group has already reported overlapping RagD mutations in patients with kidney tubulopathy and cardiomyopathy syndromes and the results displayed by the authors are not sufficient to show the role of MiT/TFE. Unfortunately, this paper seems to be less clear that the authors provide a scientific advance for publication in Nature Communications.

We would like to emphasize that, contrary to the previously published study (Schlingmann, K.P. et al. *J Am Soc Nephrol.* 2885-2899 (2021)), in our manuscript we show that pathogenic RagD mutations, including the ones reported by Schlingmann et al, do not lead to mTOR hyperactivation towards canonical substrates such as S6K, which is phosphorylated at the same levels as in normal conditions. Instead, we show that these RagD mutations induce selective phosphorylation of TFEB through a recently described non-canonical mTORC1 signaling pathway (Napolitano, G. et al. *Nature.* 585(7826):597-602 (2020); Napolitano G. et al. *Trends Cell Biol.* 32, 920–931 (2022)). TFEB phosphorylation, in turn, leads to its cytoplasmic re-localization and inhibition of its transcriptional activity. Therefore, our data indicate that mTOR hyperactivation is not the underlying mechanism causing kidney tubulopathy and cardiomyopathy syndrome, which instead is caused by TFEB inhibition, as also supported by previously reported observations in model organisms (Kim M, et al. *Int J Mol Sci.* 22(11):5494 (2021)). This is a totally novel mechanism for this disease.

Major concerns,

1. In Fig. 1E, the authors showed the different roles of RagD mutations on RagD protein. However, there were no supporting evidences in the manuscript.

To address the molecular consequences of the mutations on the structure of RagD, we predicted the effect of the identified mutations on protein stability and ligand (nucleotide) binding. These data are shown in supplementary Tables 2 and 3. In addition, we generated *in silico* models of the mutations using as a template the structure of RagD-WT GTP bound. Zoom views of these models have now been included in the new Supplementary Figure 1B-D. We would like to highlight that the predictions, all of which suggest reduced GTP binding and hydrolysis, are in line with the results from the GTP binding assays and GTPase activity assay presented in the new Supplementary Figure 1E-H, respectively. Finally, we edited the main text (see paragraph: In silico modeling and in vitro assays reveal the gain of function of RagD mutations) in the Results section of the manuscript and the new Supplementary Figure 1 legend to state the above-mentioned points as clearly as possible.

2. In Fig. 1F, GTP-bound levels of RagD mutants were shown using GTP- conjugated beads. The authors also should check GTP status of RagD mutants by enzymatic GTPase assay method.

We thank the reviewer for suggesting this type of experiment. We performed the GTPase activity assays with all RagD mutants. We were glad to see that all RagD mutants showed

negligible GTPase activity, confirming both our previous data and our structural predictions (see previous point). These new data are now shown in Supplementary Figure 1 G,H of the revised manuscript.

3. In Fig. 2, Fig. 3, Supplementary Fig.1, and Supplementary Fig.2, the authors should also check the results in control cells (FLCN WT). Furthermore, the author only checked the effect of RagD mutations in FLCN KO cells.

Our goal was to test whether RagD mutations induced TFEB phosphorylation and cytoplasmic re-localization. In untreated control cells (FLCN WT), TFEB is already phosphorylated and cytoplasmic, therefore, this would not be the right type of control for this experiment. This is the reason why in Figure 2 and Supplementary Figures 1,2, 3 of the revised manuscript we did not use FLCN WT cells. Instead, in Figure 3 and Figure 4 and in Supplementary Figure 4 of the revised manuscript we used FLCN WT cells treated with different drugs that induce TFEB nuclear translocation. Under these conditions we could test whether RagD mutants induced TFEB cytoplasmic re-localization in WT cells treated with the drugs. We hope that this clarifies the issue.

Leucyl-tRNA synthetase 1 has been known to be a RagD-GAP and the authors should also check the effects of RagD mutations in leucyl-tRNA synthetase 1 KO cells.

As requested by the reviewer, we tested the effects of the RagD mutations in the LARS1 knockdown cell line (siRNA LARS1). The results obtained in both HEK293T and HeLa cell lines show no effect on TFEB phosphorylation in the presence of RagD mutations in a LARS1 knockdown condition. Indeed, in the WBs shown in Reviewers Figure 1, the presence of two bands revealed with anti-TFEB antibody in LARS1-KD cells transfected with RagD S77L and P88L mutants clearly indicates that TFEB is phosphorylated.

Reviewers' figure 1: HEK293T (A) and HeLa (B) cells transfected for 72h with siRNA LARS1 or siRNA scramble and then transfected for 24h with plasmids encoding RagD WT or RagD S77L or P88L. Cell protein extract were immunoblotted with anti-LARS1, anti-HA, anti-TFEB and anti-Actin antibodies.

4. In Fig. 2A, the authors showed that exogenous FLAG-tagged TFEB coimmunoprecipitated with RagA/RagD heterodimer containing RagD mutations. However, this result does not reveal whether TFEB binds to RagA or RagD. The authors should check whether endogenous TFEB also binds to RagA or RagD and that this co-immunoprecipitation is mTORC1 dependent. In addition, RagD has been known to form a complex with RagB rather than RagA. What happens to TFEB binding if RagB is overexpressed instead of RagA?

We thank the reviewer for giving us the opportunity to clarify this crucial aspect of the RagD-A/TFEB interaction. We would like to point out that the IP in Figure 2A already shows that single overexpression of RagA is not sufficient to bind TFEB (first lane of WB). We also confirmed the same results by overexpressing RagB (Supplementary Figure 1J), as requested by the reviewer. As shown both in Figure 2A and in Supplementary Figure 1J overexpression of RagA or RagB alone is not sufficient to bind TFEB.

Moreover, as suggested by the reviewer, we tested whether endogenous TFEB binds to RagD mutants in *FLCN-KO* cells. As shown in Figure 2B, in *FLCN-KO* cells the RagD mutants interact with endogenous TFEB in an mTORC1-independent manner (also in presence of Torin), whereas WT RagD does not interact with TFEB.

5. In Supplementary Fig. 1A, the authors showed that RagD mutations induced TFEB phosphorylation. However, the data on the mechanisms by which RagD mutations induce TFEB phosphorylation are lacking.

We and others previously demonstrated that, unlike the phosphorylation of canonical mTORC1 substrates such as S6K, TFEB phosphorylation requires active RagC/D which mediate mTORC1 substrate recruitment for TFEB. Indeed, we and others showed that expression of active RagC (S75L) or RagD (S77L) mutants in *FLCN* knockout cells rescued TFEB phosphorylation (Napolitano, G. et al. *Nature*. 585(7826):597-602 (2020); Lawrence, R.E et al. *Science*.366(6468):971-977 (2019); Li, K. et al.. *PLoS Biol* 20(3):e3001594 (2022)).

6. In Fig. 2B, if *FLCN* is a GAP for RagC/D, p-S6K level should be reduced in *FLCN* KO cells than in WT cells. But, we cannot confirm this. Also, since there is no effect of RagD mutations on p-S6K, the results of mTORC1-activating RagD mutations cannot be confirmed.

Unfortunately, this reviewer ignores recent studies that reported a new “non canonical” mTORC1 signaling pathway that, differently from the canonical one, is dependent on *FLCN* and RagC/D activities but is independent from growth factors and Rheb activity. These studies clearly indicate that phosphorylation of S6K is independent from *FLCN* and RagC/D activities. (Napolitano, G. et al. *Nature*. 585(7826):597-602 (2020); Napolitano G. et al. *Trends Cell Biol*. 32, 920–931 (2022); Lawrence, R.E et al. *Science*.366(6468):971-977 (2019); Li, K. et al.. *PLoS Biol* 20(3):e3001594 (2022); Nakamura, S. et al. *Nat Cell Biol* 22, 1252-1263(2020).; Goodwin, J.M. et al. *Sci Adv* 7, (40) eabj2485 (2021)).

7. In Fig. 2 and Supplementary Fig.1, the authors used FLCN KO cells. The authors should check that these cells are actually FLCN KO.

As shown in Reviewers Figure 2, the FLCN-KO cell line lacks FLCN protein.

Reviewers' figure 2: Immunoblot of cell lysates of WT and FLCN-KO HeLa and HK-2 cells showing the absence of FLCN protein in both FLCN-KO cell lines.

8. In Supplementary fig. 1C, the authors showed lysosomal localization of RagD WT, S77L and P88L. What about other RagD mutants? The authors should quantify the result of Supplementary Fig. 1C.

We are afraid that the reviewer misinterpreted our data. In Supplementary Fig. 1C we did not show lysosomal localization of RagD WT, S77L and P88L. What is shown in this figure is the localization of mTOR, as it was clearly indicated in the figure legend. We have now quantified mTOR-lysosomal localization in all mutants by High Content analysis. The results are shown in Supplementary Figure 2E,F.

9. In Supplementary Fig. 1D and E, the authors should confirm whether TFEB is also localized in the lysosomes with RagD through Immunofluorescence or lysosomal fraction.

We thank the reviewer for this suggestion. As requested, we replaced the panel in Figure 2C and D with immunofluorescence showing TFEB/Lamp1 co-localization in cells overexpressing RagD mutants. These results are now present in the new Figure 2E, and G and in Supplementary Figure 3A, B.

10. The suppression of nuclear localization of TFEB by exogenous RagD mutants may be an artificial effect. It should be checked whether the cellular localization of TFEB is regulated through the introduction of endogenous RagD mutations.

We thank the reviewer for this comment. The possibility that the suppression of TFEB nuclear localization by exogenous RagD mutants is an artificial effect is very unlikely since this does not occur when overexpressing the WT form of RagD. However, to address in a more definitive manner the reviewer's criticism, we have now used fibroblasts from patients from the newly identified family with kidney tubulopathy and cardiomyopathy described in our paper, carrying the newly identified RragD P88L mutation. As shown in RagD mutants overexpression experiments, TFEB failed to translocate to the nucleus upon amino acid

starvation in fibroblasts from RRAGD P88L patients. These important, disease-relevant, new results are now shown in Figure 2 D,I,J of the revised manuscript.

11. In Fig. 3A and Supplementary Fig. 2, the author confirmed changes in TFEB nuclear translocation by LLOMe, MK6-83 and O/A treatment. So what about autophagy system? Will it recover normally?

Unfortunately, the reviewer misinterpreted our data. The evaluation of autophagy pathway in cells treated with LLOMe and MK6-83 has been extensively characterized (Nakamura, S. et al. Nat Cell Biol 22, 1252-1263(2020); Scotto Rosato A, et al. Nat Commun. 10(1):5630 (2019)) since there is a clear impact of Lysosome damaging compounds (LLOMe) and TRPML1 agonists (MK6-83) on this pathway. We have tested the possible contribution of RagD WT and P88L mutant overexpression on the autophagy pathway by analyzing the levels of LC3-II and p62 under basal and starvation (HBSS) conditions and under Bafilomycin (Baf) treatment and no alterations in autophagic flux were observed, as shown by WB experiments reported below in which p62 is clearly degraded upon autophagy induction (HBSS) and accumulated upon Baf treatment. Furthermore, we analyzed the autophagy flux in a HeLa cell line stably expressing the GFP-RFP tandem LC3, a standard tool to evaluate autophagy flux, transiently transfected with RAGD WT and P88L and again no alteration of autophagy flux was observed (see Reviewers figure 3)

Reviewers' figure 3: (A) HeLa cells transfected with plasmids encoding RagD WT or RagD P88L either untreated (BAS) or treated for 6h with HBSS or Bafilomycin A1 (Baf). Cell lysates were then subjected to immunoblotting experiment with anti-HA, anti-p62, anti-LC3 and anti-actin antibodies. (B) Quantitative analysis of autolysosome number (Spot R-Spot G) in HeLa cells stably expressing GFP-RFP tandem LC3 and transfected with plasmid encoding RagD WT or RagD P88L either untreated or treated for 6h with HBSS or Bafilomycin A1 (Baf).

12. In Fig. 3, all experiments were done in control cells (FLCN WT). What about FLCN KO cells?

Unfortunately, the experiment suggested by this reviewer is illogical. In WT cells treatment with the drugs induces TFEB nuclear translocation. We used WT cells treated with the drug to check for the induction of TFEB cytoplasmic localization by the RaD mutants. It does not make sense to treat FLCN KO cells with the drugs because TFEB is already nuclear in FLCN-KO cells.

13. In Fig. 3A, the authors expressed statistical significance between control-transfected, LLOMe-treated group and RagD WT-transfected, LLOMe-treated group. However, there appears to be no statistical significance between the two groups.

We thank the reviewer for pointing out this graphical error to us which we have corrected in the last version of the work.

14. In this manuscript, there are no supportive data on the functional importance of TFEB regulation by RagD mutations in kidney tubulopathy and cardiomyopathy syndromes.

To address this point, we differentiated cardiomyocytes (CMs) from human pluripotent stem cells (hiPSCs). This is a physiological and disease-relevant system. Human induced pluripotent (hiPSC)-CMs were transfected with either WT mRNA-RagD or mutant mRNA-RagD (S76L, P88L and T97P) and then treated with LLOMe. As shown in the new Figure 3G,H, TFEB nuclear translocation was significantly reduced in cardiomyocytes transfected with RagD mutants, thus confirming the results obtained on HeLa and HK-2 cell lines and on patient-derived fibroblasts. Furthermore, lentiviral transduction of hiPSC-CMs with WT RagD and mutant RagD (S76L and P88L) subjected to amino acid starvation for 6h showed impaired TFEB nuclear translocation only in RagD mutant-transfected cells (Supplementary Figure 5A). Finally, we would like to emphasize that the patients' primary dermal fibroblasts also show impairment of TFEB nuclear translocation (Figure 2I,J), supporting the results from hiPSC-CMs. These new data demonstrate that the pathways impaired by RagD mutations are the same in HeLa, HK-2, cardiomyocytes and patients' fibroblasts. This is an important, disease-relevant, addition to our study.

Minor concerns,

1. In supplementary Fig. 1, the author overexpressed FLAG-TFEB, but what is GFP-TFEB? We thank the reviewer for pointing out this typographical error which we have now corrected.

2. In HA-RagD blot of Fig. 3D, overexpression of RagD S77L does not appear to increase RagD expression correctly.

According to the reviewer's suggestion, we replaced the immunoblot shown in Figure 3D with another one in which the expression of the mutants is more comparable to each other. The results are now shown in Figure 4E of the revised manuscript.

Reviewer #2 (Remarks to the Author):

In this manuscript, the authors have conducted a very interesting study showing that disease-causing mutations in kidney tubulopathy and cardiomyopathy could lead to an auto-activation of RagD. These pathogenic mutants induced constitutive phosphorylation of TFEB by non-canonical mTORC1 signaling and inhibited the nuclear translocation and transcriptional activity of TFEB in cells lacking FLCN, or in cells under lysosomal damage/mitochondrial stress. In conclusion, the authors claim that inhibition of MiT-TFE transcription factors such as TFEB, but not hyperactivation mTORC1 signaling, contributes to the pathogenesis of kidney tubulopathy and cardiomyopathy.

Although there are some issues need to be addressed, this manuscript is overall promising and will be of sufficient interest to the Nat Commun readers. The manuscript contains logical experimental design and convincing cell biology and biochemical studies. The authors put effort to show the pathogenic role of RagD activation in TFEB-related disease. It will not only help to understand the basic cell biological mechanism, but also shed light to the potential therapeutic applications bases on the findings in this paper.

We thank this reviewer for the very positive comments on the value of our study.

Specific comments on the manuscript are as follows:

1. The authors claimed that “Co-immunoprecipitation (Co-IP) analysis between RagD and TFEB showed that all RagD disease-causing mutations rescued the ability of RagD to interact with TFEB... (Figure 2A)”. However, the protein levels of FLAG-TFEB in the (1%) input blot were different (e.g., S76L and T97P levels were much higher than WT). Therefore, it is recommended to re-perform the Co-IP assay using similar amount of FLAG-TFEB WT and mutants.

We thank the reviewer for this observation. We repeated this experiment and we consistently observed that the amount of both exogenous (Figure 2A) and endogenous TFEB (Figure 2B) increases upon RagD mutant transfection. We thought that this was an interesting observation. Indeed, we found that the observed increase of TFEB protein levels was due to the increased interaction with 14-3-3 which allows for its accumulation/stability. These new data are shown in Supplementary figure 2H and described in the results section: RagD auto-activating mutants inhibit TFEB/3 activity. We have also performed a Co-IP experiment in which we normalized the protein levels of TFEB, that also shows that RagD disease-causing mutations rescued the ability of RagD to interact with TFEB (see Supplementary Fig.1I in the revised manuscript).

2. There is no formal demonstration of the “lysosomal” association between RagD and TFEB – if RagD physically interacts with TFEB and thereby allows mTORC1 to phosphorylate TFEB, one may speculate that TFEB (and active RagD) can colocalize with lysosome (Lamp1).

We thank the reviewer for this suggestion. We replaced the panels in Figure 2C and D with panels showing TFEB/LAMP1 co-localization in cells overexpressing RagD WT and RagD mutants. These results are now present in the new Figure 2E, G and Supplementary Figure 3A, B.

3. The inhibition of the expression of TFEB-targeting genes in Fig 2G is convincing. However, to fully support that the inhibition relies on the regulation of TFEB, these experiments should be also done in TFEB knockdown/knockout cells (or TFEB/TFE3/MITF triple KD/KO cells).

We thank the reviewer for this suggestion. We repeated the same experiment in TFEB/TFE3 knockdown cells (siRNA TFEB/TFE3) where we confirmed that the inhibition of the expression of target genes is correlated with the expression of TFEB and TFE3. Moreover, we performed a similar experiment using a GPNMB reporter, which showed that silencing TFEB/TFE3

decreased the reporter activity. These new results are now present in Supplementary Figure 3E,G.

4. Fig 3D: the reduction of both PINK1 and Parkin protein levels is interesting. In fact the “activation” of PINK1/Parkin-mediated mitophagy does not necessarily rely on the protein level of Parkin, but requires Parkin mitochondrial translocation. Therefore, cellular localization of Parkin should be carefully examined to show mitophagy activity. In this case expression of exogenous Parkin (e.g., Parkin with a fluorescent tag or an epitope tag) could be helpful.

Following the reviewer's suggestion, we performed immunofluorescence staining of endogenous Parkin and Tomm20 in HK-2 cells overexpressing RagD WT and mutants after O/A treatment. Confocal images and graph in new Figure 4C, D show the co-localization between Parkin and mitochondria (Tomm20) in HA-positive cells, highlighting the impairment of Parkin mitochondrial translocation in cells expressing RagD mutants compared to cells expressing RagD WT construct.

5. The reviewer suggests the authors to move at least some of the Supplementary Figures to the main figures. In addition, pTFEB (S211) immunoblots need to be shown in Supplementary Fig 1B, and the quality of Supplementary Fig 2C (lower panels) need to be improved.

We thank the reviewer and have changed the figures according to the new data obtained.

6. The manuscript is overall well written, but multiple errors appear in the text and figures (e.g., Supplementary Fig 1A: if the labelings were correct, HA-RagA and FLAG-TFEB (GFP-TFEB?) were supposed to be detected on the left panels.

We thank the reviewer for pointing out these errors, which we have now corrected.

Reviewer #3 (Remarks to the Author):

“RagD autoactivating mutations impair MiT/TFE activity leading to kidney tubulopathy and cardiomyopathy” describes a novel RRAGD patient mutation, p.Pro88Leu, associated with kidney tubulopathy and cardiomyopathy. This is the second report of RRAGD-related kidney tubulopathy and cardiomyopathy syndrome, following a recent publication of Schlingmann et al.. The authors examined the structural consequence of the current known RagD mutations and further looked into the GTP-binding capability of these mutants that resulted in impaired GTP binding, demonstrating a gain of functionality. Furthermore, the authors, consistent with a previous publication by Kim et al., demonstrated that this new RagD mutation along with previously reported mutations led to impaired regulation of the MiT-TFE factors and consequent dysregulation of lysosomal/autophagic response.

Overall, this work presents interesting findings that will contribute to more understanding of the Rag GTPases functionality. However, the molecular pathogenesis of the kidney and heart disease caused by RagD mutations remains elusive and there are several contradictions with

previous literature.
Please find my comments below:

Major comments

1. The main issue of the manuscript is that the authors only provide cellular data in HeLa and HK-2 cells to study the role of RRAGD mutations. Although these experimental results are novel and interesting, they do not support claims about the molecular causes of cardiomyopathy and tubulopathy. As both mTOR hyperactivation and TFEB loss-of-function are associated with cardiomyopathy. Additional data from in vivo models would be required to answer this question. Although I understand that this may be beyond the feasibility of this revision, claims about the role of TFEB/mitophagy in the pathogenesis of the disease in the abstract and discussion section should be prevented.

We followed the reviewer's suggestion and "tuned down" the claim that inhibition of mitophagy is the underlying mechanism of the kidney tubulopathy and cardiomyopathy. Most importantly, we believe that the new data obtained in cardiomyocytes and in patients' fibroblast (see new Figure 2D,I,J, and Supplementary Figure 2G, Figure 3G,H and Supplementary Figure 5A,B) make a much stronger case supporting the role of TFEB and MiT-TFE factors in this disease.

2. The authors showed that mitophagy is indeed impaired in RagD mutants-expressing cells through reduction of Parkin and PINK1 proteins after treatment with ATP synthase inhibitor oligomycin and the complex III inhibitor antimycin A (O/A). Moreover, TFEB is retained in the cytoplasm in these cells upon O/A treatment. However, there is a lack of experiments to further demonstrate that specifically mitophagy is responsible for the kidney tubulopathy and cardiomyopathy in the patients. Are biopsies or patient cells (fibroblasts?) available that could confirm mitophagy in patients?

As explained in the response to point 1 of the same reviewer, we cannot conclude that impaired mitophagy is the main mechanism underlying kidney tubulopathy and cardiomyopathy syndrome. Therefore we "tuned down" this claim in the discussion section. However, experiments performed in *RRAGD* P88L patients' fibroblasts suggest that mitophagy is impaired as determined by the analysis of Pink, whose level does not increase upon O/A treatment as in Control (CTRL) fibroblasts (see Reviewers Figure 4).

Reviewers' figure 4: Fibroblasts from healthy donor (CTRL) and from RagD-P88L patient were either untreated (FED) or treated for 6h with Oligomycin/Antimycin (O/A). Protein extracts were analyzed by immunoblot with anti-PINK1 and anti-actin antibodies.

3. Previous studies have identified downstream targets of TFEB in cardiac dysfunction and mitochondrial signaling. Have downstream targets like GATA4 (Song et al.) or MCU and calcineurin signaling (Kenny et al.) been examined?

We thank the reviewer for pointing out the importance of the analysis of cell-specific TFEB target genes. We generated human iPSC-derived cardiomyocytes (hiPSC-CMs) and transfected them with mRNA-RagD WT or mRNA-RagD mutants (S76L and T97P) and then treated them with LLOMe for 6 hours to induce TFEB nuclear translocation and the subsequent induction of its target genes. Interestingly, qPCR analysis of GATA4, PGC1 α and MCU mRNA levels revealed that whereas GATA4, PGC1 α and MCU are upregulated in hiPSC-CMs transfected with RagD WT, their upregulation is markedly reduced in hiPSC-CMs transfected with S76L and T97P RagD mutants (see Reviewers' figure 5). These data indicate that GATA4 and MCU are actually TFEB target genes in hiPSC-CMs and that, if TFEB nuclear translocation is prevented, their level cannot be induced in response to endogenous or exogenous stimuli.

Reviewers' figure 5: Quantitative real-time PCR analysis of GATA4, PGC1A and MCU mRNA levels in hiPSCs-derived cardiomyocytes transfected with RagD-T97P or RagD-S76L normalized on RagD-WT transfected cells.

3. The authors do not demonstrate effects of previously published RRAGD mutations on p70S6K phosphorylation. This is in contrast with previously published data by Schlingmann et al. However, the experimental conditions are slightly different (no IP).

What is the percentage of RagD-transfected Hela FLCN KO cells? Could the phosphorylation be mostly dependent on non-transfected cells? Could low transfection efficiency in these cells be the reason that there were no significant changes in phosphorylation of 70S6K?

We thank the reviewer for raising this point. Although there are some slight differences in the experimental approaches, to clarify this crucial aspect we performed exactly the same experiment as in Schlingmann et al. As shown in the IP experiment below, we found no significant increase in the phosphorylation of p70S6K on residue T389 upon transfection of RagD P88L mutant compared with RagD wild type (see Reviewers Figure 6)

Reviewers' figure 6: Immunoprecipitation performed in HEK293T cells transiently transfected with FLAG-S6K (bait) to monitor the mTORC1 signaling in response to amino acids (+) or starvation (-) following the phosphorylation of S6K on Threonine 389 in presence of RagD WT or RagD mutant P88L. HA-RagD WT or RagD mutant were transfected in equimolar amount with RagA. Constitutively active RagA-Q66L was used as positive control.

Importantly, we also analyzed the phosphorylation of p70S6K in *RRAGD* P88L-derived fibroblasts under fed conditions and after amino acid starvation. Again, we found no changes in the phosphorylation of p70S6K, thus confirming our data obtained in HeLa and HK-2 cell lines (as shown in Figure 2D). We wish to emphasize, once again, that these data are in line with previous studies, including our own, indicating that RagC/D activities are essential for mTORC1-mediated TFEB phosphorylation but they are dispensable for the phosphorylation of other mTORC1 substrates such as S6K and 4EBP-1. (Napolitano, G. et al. *Nature*. 585(7826):597-602 (2020); Lawrence, R.E et al. *Science*.366(6468):971-977 (2019); Li, K. et al. *PLoS Biol* 20(3):e3001594 (2022)).

4. Along the same lines, the data demonstrating reduced lysosomal localization (Suppl Fig 1C) is in contradiction with the paper of Schlingmann et al., demonstrating increased interaction of a RagD mutant with mTOR and lysosomal V-ATPases. Could this be explained by the differences in cell type?

We thank the reviewer for this comment. The experiment in Supplementary Fig. 1C shows that in cells transfected with RagD mutants, amino acid starvation was still able to induce the release of mTOR from the lysosome. In a new set of experiments, we confirmed the same results in fibroblasts from patients carrying the *RRAGD* P88L mutation (as shown in supplementary Figure 2G). Thus, according to our results, RagD disease-causing mutations do not promote mTOR localization at the lysosome, contrary to what was previously proposed by Schlingmann et al.

5. What are the authors' conclusions regarding the role of mTORC1 phosphorylation in TFEB functionality? Figure 2B and supplementary figure 1B nicely demonstrated that mTORC1 non-canonical signalling of TFEB phosphorylation is increased when RagD mutants are present in the cells, but did not elaborate further on this. Moreover, the results on LLOME and MK6-83 treatments seem to contradict the concept of mTORC1 phosphorylation-dependent as nuclear

translocation is still not induced given that mTORC1-mediated TFEB phosphorylation was inhibited in these cells.

We thank the reviewer for this comment. The role of mTORC1-mediated phosphorylation is to inhibit TFEB nuclear translocation and transcriptional activity. This has been demonstrated by many groups and published in hundreds of papers. As reported in previous studies (Nakamura, S. et al. *Nat Cell Biol* 22, 1252-1263(2020); Goodwin, J.M. et al. *Sci Adv* 7, (40) eabj2485 (2021)), in our paper we showed that treatment with LLOMe and MK6-83 promoted TFEB de-phosphorylation and nuclear translocation without affecting mTORC1 activity towards other substrates. We also showed that transfection of RagD mutants in cells treated with LLOMe and MK6-83 promoted TFEB rephosphorylation and cytoplasmic relocalization, once again without affecting p70SK6 phosphorylation (see new Figure 3C,F).

6. In the discussion section, the authors discuss quite thoroughly the roles of mitochondria in kidney and heart and how TFEB contributes to mitochondria quality control. While this hypothesis is promising, discussion on other results shown in the manuscript is missing. The discussion section now mainly cites supportive results, it would be important to also highlight the contradictions with the literature. E.g. Kim et al. demonstrate that mTOR inhibition did not ameliorate the cardiac phenotype in zebrafish, whereas this manuscript claims that mTOR inhibition decreases TFEB phosphorylation. E.g. the above mentioned issues with the Schlingmann paper.

Following the reviewer's suggestion, we included a dedicated paragraph for the discussion and expanded this section. Concerning the putative discrepancy with Kim et al., who reported lack of amelioration of the cardiac phenotype, using rapamycin-mediated inhibition of mTORC1 activity, in a zebrafish model carrying a RagC activating mutation, we wish to clarify that we and others reported that TFEB is a Torin-sensitive but rapamycin-insensitive mTORC1 substrate (Settembre et al., *EMBO J.* 31(5):1095-108 (2012); Ferguson RA et al, *Sci Signal.* 5(228):ra42(2012)). Therefore, our data are not in contrast with those reported by Kim et al.

7. Interestingly, the authors describe the first RRAGD mutation outside of the G-boxes, could they speculate on the role of the G-boxes for the disease?

We thank the reviewer for this comment. G proteins have a conserved G domain that has the ability to bind both GTP and GDP. Interestingly, unlike most other proteins that use alpha helices and beta sheets to mediate binding to ligands, G proteins use 5 loop regions to mediate their interaction with nucleotides. G1 (P loop) and G2 (switch I) interact with the phosphate and help coordinate the magnesium ion; G3 contains the critical Gln residue responsible for GTP hydrolysis, and G4 and G5 make contacts with the guanine base helping to distinguish it from other nucleotides. Upon exchange of the nucleotide, major conformational changes take place within Switch I and Switch II that allows binding of the gamma phosphate. As such, the flexibility of the loops is required for the activation cycle of the G proteins. As described in Schlingmann KP et al, most RagD mutations that lead to its auto-activation occur in highly conserved residues that lay within these critical G boxes. Speculatively, the authors proposed that the disease variants analyzed may display low

nucleotide affinity and fast nucleotide exchange, thus interfering with RagD mediated signaling. No experimental evidence was provided to support this hypothesis. Here we confirmed the assignment of S76, T97, P119 and I221 to the G boxes of RagD. In addition, we addressed experimentally the impact of the mutations on RagD load. First, we predicted the effect of the identified mutations on protein stability and nucleotide binding (Tables 2 and 3). In addition, we generated *in silico* models of the mutations using as a template the structure of RagD GTP bound (Figure 2). We then functionally corroborated these predictions by performing GTP binding assays and GTPase activity assays as presented in Figure 1E-F. These data show that the RagD mutants are unable to bind GTP and are thus in a nucleotide-free state or a GDP-loaded state, both active conformations, suggesting that they might lead to constitutive auto-activation of RagD. Moreover, we could show that the rate of GTP hydrolysis is severely compromised in all mutations analyzed. Together, our results strengthen the importance of residues within G boxes for the regulation of the respective G proteins. Due to their unique structural properties, with a rotationally constrained rigid-ring structure, prolines are very particular residues. P88 is present between the G1 (p-Loop) and the G2 (Switch I). As such, it is tempting to propose that the introduced restraint in the backbone helps orient G1 and G2 to facilitate magnesium coordination. Mutations to another residue would most probably interfere either with the rearrangements of G2 (Switch I) occurring during the activation cycle or most probably, based on our predictions, an incorrect orientation between G1 and G2 that interferes with protein stability when in the GTP bound state. The consequence of this mutation is a drastic change in nucleotide preference with RagDP88L found exclusively bound to GDP. Overall, the P88L mutation highlights the importance of the inter G box spacer regions not only in defining the length between the loops but also in providing the necessary three-dimensional orientation of the loops for adequate nucleotide binding. The above considerations have been added to both results and discussion (albeit in a shorter format).

Minor comments

1. Please check throughout the manuscript the correct use of gene names *RRAGD/RRAGC* (italics) and protein names RagC/RagD.

We thank the reviewer for this comment. We corrected these errors in the new version of the manuscript.

2. Figure 2B a. What could be the reason that mutations on the S76 and S77 sites resulted in higher TFEB levels compared to the other mutations?

We believe that this difference derives from a technical issue during WB acquisition. Therefore, we repeated the experiment and found comparable levels of TFEB in cells transfected with S76L and S77L mutants (new Figure 2C).

b. In Figure 2B and Supplementary fig 1B, molecular weight shift in TFEB is considered as change in phosphorylation. However, pTFEB (S211) antibody was used in Supplementary Figure 1A. Could the authors comment on the different detection methods between these?

Unfortunately, phospho-antibodies against pS211 have the sensitivity to detect p-TFEB only in cells in which TFEB is overexpressed, whereas they are unable to detect endogenous p-TFEB. Therefore, in Figure 2B (now Supplementary figure 2A) and Supplementary Fig 1B (now Figure 2C), phosphorylation of endogenous TFEB was evaluated by analyzing the molecular weight shift of TFEB in WB experiments, whereas in Supplementary Figure 1A (now Supplementary figure 2C) phosphorylation of overexpressed TFEB-GFP was analyzed by using pS211 phospho-antibodies.

3. Figure 2E

a. Typo in the legend (rapresentative should be representative)

We thank the reviewer for pointing out this typographical error which we have now corrected.

b. What is the basal fluorescent of the reporter in non-transfected cells?

In this experiment we used a HeLa FLCN KO cell line, in which TFEB is constitutively nuclear and active, stably expressing a TFEB transcriptional reporter. This cell line was selected with puromycin and sorted for the mCherry signal.

4. Figure 2F

a. What does “% of PE fluorescent” exactly represent? Intensity of the fluorophore? Was this normalised to anything?

“% of PE fluorescent” indicates the percentage of positive fluorescent events calculating the ratio between the positive events of PE signal on the HA-positive cells.

5. Figure 3A, B, and C:

a. Images shown should be representative of the quantification graphs shown rather than effects of drug treatments in WT cells.

We thank the reviewer and we apologize for the confusion. In the new version of the manuscript we present in a clearer way the images with the relevant quantification graphs.

6. Figure 3D:

a. It is not really clear what is quantified here. Please add the y-axis titles to the graphs

We apologize if we were not clear enough in Figure 3D (now Figure 4F, G).

Previously the title of y-axis box was shown above the related graphs. In the new version of Figure 4 we followed the reviewer’s suggestion.

7. Supplementary Figure 1A:

a. Typo mutation name S211K should be I221K

We thank the reviewer for pointing out this typographical error which we have now corrected.

b. In the figure, FLAG-TFEB was used for transfection but GFP-TFEB was used to label the blot. Is this correct?

We thank the reviewer for pointing out this typographical error which we have now corrected.

c. In RagD empty condition, TFEB and RagA are also absent. Was this perhaps a triple empty condition?

Yes, indeed it is a triple empty condition.

Reviewer #4 (Remarks to the Author):

Irene Sambri et al described that a novel mutation in RRAGD can lead to kidney tubulopathy and cardiomyopathy via inhibiting MiT-TFE factors but not mTORC1 hyperactivation. The study is interesting but is insufficient for NC at present.

major issues:

1. All the mutations studies was conducted in Hela cells with plasmids transfection, which can not prove the viewpoint proposed by the authors. I recommend the authors to construct the mutant kidney cell line and cardiomyocytes by Cas9.

We thank the reviewer for this comment. We agree on the importance of more physiological cell models, thus we added experiments using human cells. Unfortunately, the relevant human cells – cardiomyocytes - are a non-dividing cell type and for this reason CRISPR/Cas9 technology is not applicable to insert mutations in this cellular model. However, we used fibroblasts from patients carrying RRAGD P88L mutation to show the effects of endogenous RagD mutants. We also transfected hiPSC-derived cardiomyocytes with RagD WT and mutants (S76L, P88L and T97P) to validate our results on a more disease-relevant and physiological cell type (Figure 3 G,H and Supplementary Figure 5A), see also response n 14 to reviewer 1. All the results obtained on both hiPSC-derived cardiomyocytes and patient's fibroblasts fully confirmed and robustly validated what we observed in HK2 and HeLa cells transiently transfected with RagD mutants.

2. The study need more evidence to prove the viewpoint in animal models such as point mutation knock in mice.

We thank the reviewer for this comment. While we agree that studying an *in vivo* model could give us more insight, this is currently beyond the scope of this work as it would take a very long time. On the other hand, as mentioned above, in this new version of the manuscript we analyzed the effect of RRAGD mutations both in P88L patient-derived fibroblasts and in hiPSC-derived cardiomyocytes expressing RagD mutants, which represent a very relevant and physiological way to prove our hypothesis. The new results are now present in Figure 2I-J, supplementary figure 2G ,Figure 3 G,H and supplementary figure 5A,B of the revised manuscript.

3. The genetic relationship between RRAGD and kidney tubulopathy and cardiomyopathy have been reported in J Am Soc Nephrol. 2021 Nov;32(11):2885-2899. I recommend to detect more other novel mutations.

We are very surprised by this comment of reviewer. Kidney tubulopathy and cardiomyopathy is an extremely rare condition, with only 9 families described in the literature (Schlingmann, K.P. et al. mTOR-Activating Mutations in RRAGD Are Causative for Kidney Tubulopathy and

Cardiomyopathy. J Am Soc Nephrol 32, 2885-2899). We were extremely lucky to identify the 10th family thanks to the help of an expert nephrologist who is a coauthor in the present paper (F.T.). Asking us to identify additional families appears as a “mission impossible” request.

mirror issues:

1. the pedigree of the family may have some problems, such as II-3,II-4, II-5 and II-6.

We thank the reviewer for pointing out these errors, which are corrected in Figure 1 of the new version of the manuscript.

2. several data were showed in mRNA levels which is not enough, please provide the WB data.

We thank the reviewer for this suggestion. To address this point, we performed several WB experiments. In all experiments, the WB data confirmed the results obtained with mRNA (see Supplementary Figures 4 F and G).

REVIEWER COMMENTS

Reviewer #1 (Remarks to the Author):

Authors reported the novel RRAGD mutation (p.Pro88Leu) in a family with kidney tubulopathy and cardiomyopathy syndrome. In addition to the previously described mutations reported by Schlingmann et al., authors have demonstrated all RRAGD mutants associated with upper mentioned syndromes, have reduced GTP binding ability and led to inhibition of mTORC1 factors. Authors have seriously overviewed their previous work, went through all the reviewers' comments and made considered replies. This updated version of the manuscript is generally well written and covers interesting findings that will improve our understandings in the pathological associations of Rag GTPases. However, there are still major concerns regarding the contradictory results with the work by Schlingmann et al. and there are some questions raised from the newly added information. Although authors clearly demonstrated that TFEB functionality is affected by the RagD mutants, the current results require more evidences to prove the substrate specific effect of RagD mutants over other mTORC1 substrates.

Major comments:

1. In the manuscript, authors described that the expression of RagD mutants enhanced the protein levels of both exogenous and endogenous TFEB, which is due to increased stability of TFEB by interacting with 14-3-3. However, this increment in TFEB level seems to occur inconsistently throughout the manuscript. For example, in figure 2D and 3C. Is this due to different cell line used in the experiment?

2. It is well known that active Rag heterodimer, consisting with GTP-bound RagA/B and GDP-bound RagC/D, recruits mTORC1 to lysosomes and drives the activation of mTORC1. Please check the major status of RagA/B that formed heterodimer with RagD mutant whether they are bound with GTP or GDP. Would RagD mutants upregulate GTP loading of RagA/B and promote mTORC1 activation?

3. Figure 2D seems to be one of the key data that shows the phosphorylation status of TFEB in human fibroblast carrying RRAGD P88L mutation. However, due to the quality of WB blot of TFEB, it is difficult to tell whether there the band has shifted or not. Same problem also appeared in some of the TFEB blots in the figures for reviewers as well. These blots should be adjusted and it would be helpful if authors show the results using phospho-TFEB antibody used in the supplementary figure 2C.

4. In figure 2D, authors showed that RRAGD_P88L overexpression in human fibroblast did not affect p-S6K levels. However, it seemed that under amino acid starvation, phosphorylation of S6K is repressed yet the phosphorylation of TFEB is unaffected when RRAGD_P88L is overexpressed (compared to the control band). Since TFEB phosphorylation is also mediated by mTORC1 which is inhibited by Torin in figure 2C, authors need to explain in more detail about this difference. Also consider to examine the difference in mTORC1 activity and the phosphorylation status of mTOR in control and RRAGD_P88L.

5. Throughout this article, authors demonstrated that RagD mutants only affects phosphorylation of TFEB but not other mTORC1 substrates. However according to the previously published data by Schlingmann et al., RagD mutants other than RagD P88L increased p70S6K levels in the presence of amino acids. In fact, previous studies by many groups using RagD mutant (S76L, S77L) also showed downstream activation of mTORC1 and increased S6K phosphorylation (Oshiro et al., JBC, 289(5): 2658-2674, 2014). Apart from the difference of the experimental conditions, what is the main reason for this difference? Are there any chances that these RagD mutants found in the patients are able to affect both TFEB and S6K?

Also, in Reviewers' figure 6, authors performed the same experiment as in Schlingmann et al. using RagD P88L and claimed there was no significant differences in p70S6K upon transfection of RagD P88L mutant compared to wild type. However, from the p-S6K (T389) blot in this figure, it seems that the intensity of bands in RagD P88L/RagA WT are in fact higher than WT/WT both with and without amino acids. Therefore, this result does not support authors' statements. Moreover, it would be better if the authors showed the results of other mutants as well in this experiment condition.

6. In reviewers' figure 1, authors stated that there was no effect on TFEB phosphorylation in the presence of RagD mutations in LARS1 knockdown condition. Previously, effect of RagD mutations in LARS1 KO cells was questioned because other than FLCN, LARS1 is also known to have GAP

activity against RagD. LARS1 knockdown was shown to increase GTP-bound RagD and decreased S6K phosphorylation (Lee et al., PNAS, 2018). Therefore, please also show the effect of RagD mutants on the phosphorylation of other mTORC1 substrates p70S6K and 4EBP1 in LARS1 KD cells as well.

7. Besides TFEB and TFE3, ULK1 also has crucial role in the regulation of autophagy and is known as mTORC1 substrate as well. What would be the effect of RagD mutants on ULK1 and autophagy?

8. The authors found RRAGD mutations in a family with kidney tubulopathy and cardiomyopathy syndrome. However, the etiology or the molecular pathogenesis of this disease caused by RRAGD mutations seems to be preliminary and remains elusive.

Minor comments:

1. In figure 4C, it would be helpful to add the labels on the left side of the panels similar to 4A.

2. In figure 4E and 4F, although quantification of PINK/Actin showed decreased levels upon O/A treatment in RagD S77L and P88L transfected cells, the WB data (PINK1 blot) does not seem to differ from empty and WT.

3. In figure 4F and 4G in all "RagD p88L", change p to capital letters.

Reviewer #2 (Remarks to the Author):

The authors have done very good job to address most of the reviewers' previous concerns, and the revised manuscript has been significantly improved. However, there is one remaining issue need to be addressed before publication, concerning the new data in Figures 4C and 4D – in general it's not easy to perform immunofluorescence experiment using three different antibodies – if the authors used goat anti-Tomm20/Parkin/HA primary antibody (suitable for the donkey anti-goat IgG (H + L) Alexa Fluor 647 secondary antibody) along with the mouse and rabbit antibodies, they should include this information in the Method or Figure Legend section in the manuscript. Otherwise it is recommended to re-perform this experiment using fluorescent tagged Parkin with immunofluorescent staining using two antibodies.

Reviewer #3 (Remarks to the Author):

I thank the authors for answering all my comments and providing additional data, demonstrating the effects of the novel RagD mutation in cardiomyocytes and patient-derived fibroblasts. These additional experiments have significantly improved the manuscript.

The main remaining issue is that the outcomes of the experiments are opposed to the results of Schlingmann et al. (JASN 2021). Whereas Schlingmann et al. demonstrate mTOR hyperactivation and lysosomal localisation in two independent cell models, the current paper demonstrates the opposite. Although I trust that both Schlingmann et al. and the authors of the current manuscript have both performed rigorous experiments, I think that this discrepancy should at least be discussed in the current manuscript.

Potentially, the mTOR hyperactivation is dependent on additional factors related to the cell model or the experimental conditions, which could provide biological insights in the role of mTOR in the disease. As the discrepancy is completely ignored in the discussion section of the current version, I would like to ask the authors to discuss potential explanations in the manuscript.

Reviewer #4 (Remarks to the Author):

except the animal validation, all the questions were addressed well.

REVIEWERS'

COMMENTS

Reviewer #1 (Remarks to the Author): Authors reported the novel RRAGD mutation (p.Pro88Leu) in a family with kidney tubulopathy and cardiomyopathy syndrome. In addition to the previously described mutations reported by Schlingmann et al., authors have demonstrated all RRAGD mutants associated with upper syndromes, have reduced GTP binding ability and led to inhibition of MiT-TFE factors. Authors have seriously overviewed their previous work, went through all the reviewers' comments and made considered replies. This updated version of the manuscript is generally well written and covers interesting findings that will improve our understandings in the pathological associations of Rag GTPases. However, there are still major concerns regarding the contradictory results with the work by Schlingmann et al. and there are some questions raised from the newly added information. Although authors clearly demonstrated that TFEB functionality is affected by the RagD mutants, the current results require more evidences to prove the substrate specific effect of RagD mutants over other mTORC1 substrates.

Major comments:

1. In the manuscript, authors described that the expression of RagD mutants enhanced the protein levels of both exogenous and endogenous TFEB, which is due to increased stability of TFEB by interacting with 14-3-3. However, this increment in TFEB level seems to occur inconsistently throughout the manuscript. For example, in figure 2D and 3C. Is this due to different cell line used in the experiment?

We thank the reviewer for this observation. Indeed we observed TFEB increased stability in HeLa cells transfected with RagD mutants, whereas such increase appeared undetectable in patient-derived fibroblast carrying endogenous *RRAGD* P88L heterozygous mutation. This is likely due to the different experimental conditions used between HeLa and fibroblasts. We added a sentence in the results section "RagD auto-activating mutants inhibit TFEB/3 activity" discussing these differences.

2. It is well known that active Rag heterodimer, consisting with GTP-bound RagA/B and GDP-bound RagC/D, recruits mTORC1 to lysosomes and drives the activation of mTORC1. Please check the major status of RagA/B that formed heterodimer with RagD mutant whether they are bound with GTP or GDP. Would RagD mutants upregulate GTP loading of RagA/B and promote mTORC1 activation?

Suppl Figure 2E shows that in the presence of RagD mutants starvation is still able to induce mTORC1 release from the lysosomal membrane, clearly indicating that they do not influence the activation of RagA/B.

3. Figure 2D seems to be one of the key data that shows the phosphorylation status of TFEB in human fibroblast carrying RRAGD P88L mutation. However, due to the quality of WB blot of TFEB, it is difficult to tell whether there the band has shifted or not. Same problem also appeared in some of the TFEB blots in the figures for reviewers as well. These blots should be adjusted and it would be helpful if authors show the results using phospho-TFEB antibody used in the supplementary figure 2C.

We disagree with this assessment of reviewer1. The WB of Figure 2D (shown on the left) is of high quality. In this WB TFEB molecular weight shift is clearly visible, as shown by the position of the bands with respect to the dashed line. Concerning the phospho-antibodies against pS211, they are unable to detect endogenous p-TFEB. These antibodies only work on overexpressed TFEB.

4. In figure 2D, authors showed that RRAGD_P88L overexpression in human fibroblast did not affect p-S6K levels. However, it seemed that under amino acid starvation, phosphorylation of S6K is repressed yet the phosphorylation of TFEB is unaffected when RRAGD_P88L is overexpressed (compared to the control band). Since TFEB phosphorylation is also mediated by mTORC1 which is inhibited by Torin in figure 2C, authors need to explain in more detail about this difference. Also consider to examine the difference in mTORC1 activity and the phosphorylation status of mTOR in control and RRAGD_P88L.

First of all we would like to point out that RRAGD_P88L was not overexpressed, as these are patient's fibroblasts.

Unfortunately, in spite of our clarification in our first point-by-point response, the reviewer continues to either ignore, or disregard, the concept of mTORC1 substrate specificity. We would like to reiterate that several groups, including ours, showed that the activity of RagC/D is needed only for the phosphorylation of TFEB but is dispensable for the phosphorylation of S6K (Napolitano, G. et al. Nature. 585:597-602, 2020; Napolitano G. et al. Trends Cell Biol. 32, 920–931, 2022; Lawrence, R.E et al. Science.366(6468):971-977, 2019; Li, K. et al.. PLoS Biol 20:e3001594, 2022; Nakamura, S. et al. Nat Cell Biol 22, 1252-1263, 2020; Goodwin, J.M. et al. Sci Adv 7, (40) eabj2485, 2021). Remarkably, in collaboration with the groups of James Hurley and Lukas Huber, we recently provided structural evidence of mTORC1 substrate specificity (Cui et al., Nature, 614:572-579, 2023).

5. Throughout this article, authors demonstrated that RagD mutants only affects phosphorylation of TFEB but not other mTORC1 substrates. However according to the previously published data by Schlingmann et al., RagD mutants other than RagD P88L increased p70S6K levels in the presence of amino acids. In fact, previous studies by many groups using RagD mutant (S76L, S77L) also showed downstream activation of mTORC1 and increased S6K phosphorylation (Oshiro et al., JBC, 289(5): 2658-2674, 2014). Apart from the difference of the experimental conditions, what is the main reason for this difference? Are there any chances that these RagD mutants found in the patients are able to affect both TFEB and S6K?

Once again, by asking this question the reviewer continues to disregard the concept of mTORC1 substrate specificity (see our responses to previous points) as published in the following papers: (Napolitano, G. et al. Nature. 585(7826):597-602 (2020); Napolitano G. et al. Trends Cell Biol. 32, 920–931 (2022); Lawrence, R.E et al. Science.366(6468):971-977 (2019); Li, K. et al.. PLoS Biol 20(3):e3001594 (2022); Nakamura, S. et al. Nat Cell Biol 22,

1252-1263(2020).; Goodwin, J.M. et al. *Sci Adv* 7, (40) eabj2485 (2021), Cui et al., *Nature*, 614:572-579, (2023)).

Also, in Reviewers' figure 6, authors performed the same experiment as in Schlingmann et al. using RagD P88L and claimed there was no significant differences in p70S6K upon transfection of RagD P88L mutant compared to wild type. However, from the p-S6K (T389) blot in this figure, it seems that the intensity of bands in RagD P88L/RagA WT are in fact higher than WT/WT both with and without amino acids. Therefore, this result dose not supports authors' statements. Moreover, it would be better if the authors showed the results of other mutants as well in this experiment condition.

We would like to clarify that our results are not in contrast with those published in the Schlingmann et al. paper, instead they are in contrast with their interpretation of the results. Figure 4B of the Schlingmann et al. paper (see below) clearly showed that the phospho-rylation of S6K in the samples transfected with the active RagD mutants is turned off by starvation just like in the control sample (see red arrows), whereas in the sample transfected with the active RagA mutant shows constitutive mTORC1 activity during starvation. These results indicate that RagD mutants have marginal effects on mTORC1-mediated phosphorylation of S6K. Finally, we would like to cite a sentence in the discussion

of the Schlingmann et al. paper: "Of note, not all identified RAGD mutations overactivated mTORC1 signaling equally. Our experiments indicate that p.Thr97Pro results in less S6K1 activation (Figure 4)...". This sentence suggests that the authors themselves casted some doubts on the ability of the mutants to induce activation of mTORC1.

Figure 4B from Schlingmann, K.P. et al. *J Am Soc Nephrol*. 2885-2899 (2021)

6. In reviewers' figure 1, authors stated that there was no effect on TFEB phosphorylation in the presence of RagD mutations in LARS1 knockdown condition. Previously, effect of RagD mutations in LARS1 KO cells was questioned because other than FLCN, LARS1 is also known to has GAP activity against RagD. LARS1 knockdown was shown to increase GTP-bound RagD and decreased S6K phosphorylation (Lee et al., *PNAS*, 2018). Therefore, please also show the effect of RagD mutants on the phosphorylation of other mTORC1 substrates p70S6K and 4EBP1 in LARS1 KD cells as well.

As already shown in Reviewers' figure 1 of the previous point-by-point response, we tested the effects of the RagD mutations in the LARS1 knockdown cell line (siRNA LARS1). The result of this experiment showed no effect of the LARS1 knockdown on TFEB

phosphorylation. Considering that this negative result will not be included in the manuscript, we do not see the point of testing the phosphorylation of S6K and 4EBP1.

7. Besides TFEB and TFE3, ULK1 also has crucial role in the regulation of autophagy and is known as mTORC1 substrate as well. What would be the effect of RagD mutants on ULK1 and autophagy?

In the previous point-by-point (point 11) we clearly showed that RagD mutations had no effect on the autophagic flux. Indeed the TFEB-mediated response to lysosomal damage MK6-83 and O/A treatment is independent from ULK1. Therefore, we do not see the point of testing the phosphorylation of ULK1.

8. The authors found RRAGD mutations in a family with kidney tubulopathy and cardiomyopathy syndrome. However, the etiology or the molecular pathogenesis of this disease caused by RRAGD mutations seems to be preliminary and remains elusive.

The data contained in our manuscript strongly suggest that kidney tubulopathy and cardiomyopathy syndrome is caused by TFEB inhibition, rather than mTORC1 hyperactivation. This is a totally novel mechanism for this disease and is supported not only by the data contained in our manuscript, but also by in line with previously reported observations in model organisms (Kim M, et al. Int J Mol Sci. 22:5494 (2021).

Minor comments: 1. In figure 4C, it would be helpful to add the labels on the left side of the panels similar to 4A.

We have added the labels to figure 4C.

2. In figure 4E and 4F, although quantification of PINK/Actin showed decreased levels upon O/A treatment in RagD S77L and P88L transfected cells, the WB data (PINK1 blot) does not seem to differ from empty and WT.

We disagree with the reviewer's assessment. Decreased levels of PINK1 are clearly visible in the WB of Figure 4E. Furthermore, the WB quantification performed on three independent experiments clearly demonstrates a significant decrease in the intensity of PINK1 bands, as shown in the graph of Figure 4F.

3. In figure 4F and 4G in all "RagD p88L", change p to capital letters.

We changed p to P.

Reviewer #2 (Remarks to the Author):

The authors have done very good job to address most of the reviewers' previous concerns, and the revised manuscript has been significantly improved. However, there is one remaining issue need to be addressed before publication, concerning the new data in Figures 4C and 4D – in general it's not easy to perform immunofluorescence experiment using three different antibodies – if the authors used goat anti-Tomm20/Parkin/HA primary antibody (suitable for the donkey anti-goat IgG (H + L) Alexa Fluor 647 secondary antibody) along with the mouse and rabbit antibodies, they should include this information in the

Method or Figure Legend section in the manuscript. Otherwise it is recommended to re-perform this experiment using fluorescent tagged Parkin with immunofluorescent staining using two antibodies.

We thank the reviewer for the positive comments. We apologize for the unclear description of the experiment performed in Figure 4C and 4D. In the new version of the manuscript we described in detail the immunofluorescence performed using anti-Tomm20/Parkin/HA primary antibody and the relative quantification. We added the information about the antibodies used in the Methods section of the manuscript and in the relative figure legend to state the above mentioned points as clearly as possible. It now reads as follows: (C) Representative immunofluorescence images of HK-2 cells transfected with HA-RagD WT or mutants (S77L or P88L) and treated with 10 μg/ml Oligomycin and 5 μg/ml Antimycin A (O/A) for 6 hours. Cells were stained with anti-Parkin (Rabbit polyclonal) and anti-HA (Rat monoclonal). Staining for Tomm20 (Mouse monoclonal) was used as a mitochondrial marker. Scale bar, 10 μm. (D) Graph shows the mitochondria-Parkin co-localization of the HA positive (RagD WT or mutants S77L, P88L) cells treated with O/A (mean ± s.d. for N=3 independent experiments, n=195 cells).

Reviewer #3 (Remarks to the Author):

I thank the authors for answering all my comments and providing additional data, demonstrating the effects of the novel RagD mutation in cardiomyocytes and patient-derived fibroblasts. These additional experiments have significantly improved the manuscript.

The main remaining issue is that the outcomes of the experiments are opposed to the results of Schlingmann et al. (JASN 2021). Whereas Schlingmann et al. demonstrate mTOR hyperactivation and lysosomal localisation in two independent cell models, the current paper demonstrates the opposite. Although I trust that both Schlingmann et al. and the authors of the current manuscript have both performed rigorous experiments, I think that this discrepancy should at least be discussed in the current manuscript. Potentially, the mTOR hyperactivation is dependent on additional factors related to the cell model or the experimental conditions, which could provide biological insights in the role of mTOR in the disease. As the discrepancy is completely ignored in the discussion section of the current version, I would like to ask the authors to discuss potential explanations in the manuscript.

We thank the reviewer for the positive comments. Concerning the discrepancy with the Schlingmann et al. paper, as also discussed in the response to reviewer 1, we would like to emphasize that indeed our results are not in contrast with those published in the Schlingmann et al. paper, instead they are in contrast with their interpretation of the results. Figure 4B of the Schlingmann et al. paper (see below) clearly showed that the

phosphorylation of S6K1 in the samples transfected with the active RagD mutants is turned off by starvation just like in the control

sample, whereas in the sample transfected with the active RagA mutant starvation had no effect. These results indicate that RagD mutants have marginal effects on mTORC1-mediated phosphorylation of S6K. Finally, we would like to cite a sentence in the discussion of the Schlingmann et al. paper: "Of note, not all identified RRAGD mutations overactivated mTORC1 signaling equally. Our experiments indicate that p.Thr97Pro results in less S6K1activation (Figure 4)...". This sentence suggests that the authors themselves casted some doubts on the ability of the mutants to induce activation of mTORC1.

Figure 4B from Schlingmann, K.P. et al. J Am Soc Nephrol. 2885-2899 (2021)

We also would like to reiterate that the ability of RagD mutants to induce phosphorylation of TFEB without affecting the phosphorylation of S6K is in line with recent studies, including our own, that demonstrate the presence of an mTORC1 substrate specific pathway, which we named "non Canonical mTORC1 signaling (Napolitano, G. et al. Nature. 585:597-602, 2020; Napolitano G. et al. Trends Cell Biol. 32, 920–931, 2022; Lawrence, R.E et al. Science.366(6468):971-977, 2019; Li, K. et al.. PLoS Biol 20:e3001594, 2022; Nakamura, S. et al. Nat Cell Biol 22, 1252-1263, 2020; Goodwin, J.M. et al. Sci Adv 7, (40) eabj2485, 2021.), Remarkably, in collaboration with the groups of James Hurley and Lukas Huber, we recently provided structural evidence of mTORC1 substrate specificity (Cui et al., Nature, 614:572-579, 2023).

In spite of these considerations, we do agree with the reviewer on the need to discuss the discrepancy with the Schlingmann et al. study. The following paragraph has now been added to the discussion of our paper: "In the original study in which kidney tubulopathy and cardiomyopathy was first described by Schlingmann et al the authors claimed that the disease was caused by mTORC1 hyperactivation induced by *RRAGD* mutations (REF). However, in that study the degree of mTORC1 hyperactivity induced by *RRAGD* mutations (as measured by S6K phosphorylation) was marginal and the effects were highly variable among the different mutations. Most importantly, the phosphorylation of S6K in samples transfected with RagD mutants was still turned off by starvation, similarly to control sample, whereas in the sample transfected with active RagA mutant starvation had no effect (see Figure 4B of reference³). These results suggest that RagD mutants do not cause constitutive mTORC1-mediated phosphorylation of S6K. Consistent with these findings, in the present study we show that RagD mutants induce constitutive phosphorylation of TFEB and other MiT-TFE factors, whereas they have no effect on mTORC1-mediated phosphorylation of S6K. The ability of RagD mutants to induce phosphorylation of TFEB without affecting the phosphorylation of S6K is in line with recent studies, including our own, that demonstrated the presence of an mTORC1 substrate-specific pathway, which we named "non-canonical mTORC1 signaling" (Napolitano, G. et al. Nature. 585:597-602, 2020; Napolitano G. et al. Trends Cell Biol. 32, 920–931, 2022; Lawrence, R.E et al. Science.366(6468):971-977, 2019; Li, K. et al.. PLoS Biol 20:e3001594, 2022; Nakamura, S. et al. Nat Cell Biol 22, 1252-1263, 2020; Goodwin, J.M. et al. Sci Adv 7, (40) eabj2485, 2021.). Remarkably, in a recent study we provided structural evidence of mTORC1 substrate specificity (Cui et al., Nature, 614:572-579, 2023)."

Reviewer #4 (Remarks to the Author): except the animal validation, all the questions were addressed well.

We thank the reviewer for the positive comment.